# High-throughput Pore-C reveals the single-allele topology and cell type-specificity of 3D genome folding

Jia-Yong Zhong [1,6], Longjian Niu [2,3,6], Zhuo-Bin Lin [4], Xin Bai[1], Ying Chen[1], Feng Luo [5], Chunhui Hou [2] ✉ & Chuan-Le Xiao [1] ✉

Canonical three-dimensional (3D) genome structures represent the ensemble average of pairwise chromatin interactions but not the single-allele topologies in populations of cells. Recently developed Pore-C can capture multiway chromatin contacts that reflect regional topologies of single chromosomes. By carrying out high-throughput Pore-C, we reveal extensive but regionally restricted clusters of single-allele topologies that aggregate into canonical 3D genome structures in two human cell types. We show that fragments in multi-contact reads generally coexist in the same TAD. In contrast, a concurrent significant proportion of multi-contact reads span multiple compartments of the same chromatin type over megabase distances. Synergistic chromatin looping between multiple sites in multi-contact reads is rare compared to pairwise interactions. Interestingly, the single-allele topology clusters are cell type-specific even inside highly conserved TADs in different types of cells. In summary, HiPore-C enables global characterization of single-allele topologies at an unprecedented depth to reveal elusive genome folding principles.

Metazoan genomes are folded into hierarchical three-dimensional (3D) structures that regulate gene expression to specify cell identity[1,2]. These structures include chromosome territories[3–6] that can be further segregated into A/B compartments (active/inactive chromatin)[7–9], topologically associating domains (TADs)[10–13], and chromatin loops[14–16]. TADs may confine regulatory activities, and disruption of TAD borders leads to developmental disorders and even tumorigenesis[17–23]. However, chromatin loops can bridge interactions between enhancers and promoters or between CTCF sites to mediate direct regulatory or structural functions[24–30].

The discovery of canonical 3D genome structures has been mainly driven by the invention of chromosome conformation capture (3C)[31] and its derivative methods, such as 4Cs[32,33], 5C[34], Hi-C[7], and other forms of high-throughput techniques that capture pairwise DNA sequences

that are physically proximal in the nuclear space[3,35–42]. Despite the tremendous advancements achieved, however, 3C-based methods can capture only pairwise interactions reflecting neither synergistic multilocus interactions nor single-allele topology in a cell population[43]. Moreover, genome structures change dynamically throughout the cell cycle[44–46] and during development and differentiation[19,24,47–51], reflect progressive transitions between biological states, and correlate with gene regulation that frequently involves multiway chromatin interactions between enhancers and promoters[27]. To fully understand the mechanisms of dynamic genome folding and functional relevance, it is critical to acquire single-allele topology in populations of cells.

Theoretically, multiway interactions between fragments in a single read can be used to identify synergistic interactions directly and to acquire single-allele topology in a cell population. A few methods that

[1]State Key Laboratory of Ophthalmology, Zhongshan Ophthalmic Center, Sun Yat-Sen University, Guangdong Provincial Key Laboratory of Ophthalmology and Visual Science, Guangzhou 510060, China. [2]State Key Laboratory of Genetic Resources and Evolution, Kunming Institute of Zoology, Chinese Academy of Sciences, Kunming 650201, China. [3]School of Public Health and Emergency Management, Southern University of Science and Technology, Shenzhen 518055, China. [4]Zhongshan School of Medicine, Sun Yat-sen University, Guangzhou 510080, China. [5]School of Computing, Clemson University, Clemson, SC 29634-0974, USA. [6]These authors contributed equally: Jia-Yong Zhong, Longjian Niu. ✉e-mail: houchunhui@mail.kiz.ac.cn; xiaochuanle@126.com

generate multiway chromatin contacts have been developed, including genome architecture mapping (GAM)[52], ChIA-drop[53], split-pool recognition of interactions by tag extension (SPRITE)[6,54], Tri-C[55], multi-contact 4C[43], concatemer ligation assay (COLA)[56], and Pore-C[57]. Among these methods, Pore-C stands out because it can capture global high-order multiway contacts, is technically simple, and captures DNA methylation simultaneously in a cell population. Because multiway contacts reflect synergistic chromatin interactions rather than multiple mutually exclusive interactions of different alleles, we can use Pore-C to reveal single-allele topology within designated genomic regions in populations of cells.

In this work, we optimized the Pore-C protocol to achieve high-throughput long-read multiway contact nanopore sequencing and developed the MapPore-C pipeline to solve the low base-calling accuracy problem. By applying high-throughput Pore-C (HiPore-C) to human GM12878 and K562 cells, we reveal an unexpected relationship between allele-specific topology and canonical 3D genome structures.

## Results

### Solving nanopore-clogging increases the output of multiway contact sequencing

The average Pore-C throughput is relatively low (Fig. 1a and Supplementary Table 1), and more expensive than traditional Hi-C for generating the same number of pairwise contacts (Fig. 1b and Supplementary Table 2), limiting its power to reveal a multiway interaction network and single-allele topology in a cell population. Despite an average 60% increase in sequencing output resulted from the improved flow cell quality, the Pore-C sequencing output is well below the whole genome sequencing, suggesting that there is much room for improvement in the Pore-C protocol. It is known that DNA-bound proteins (as small as 2 kD) can clog sequencing pores[58]. We suspected that incomplete removal of proteins crosslinked to DNA during Pore-C concatemer library preparation causes the clogging (Supplementary Fig. 1a). To solve this problem, we tested different temperatures and durations of proteinase K digestion (Fig. 1c). The purified DNAs were sequenced on the Oxford Nanopore Technology (ONT) MinION platform, and the sequencing output was increased (Supplementary Fig. 1b and Supplementary Table 3). However, the number of active pores dropped faster than in genome sequencing (Fig. 1d). Nevertheless, we confirmed that higher temperatures and longer incubation times improved the sequencing output. Using optimized conditions, we achieved an output per ONT PromethION sequencing cell (Supplementary Fig. 1d and Supplementary Table 4) ~80 Gbase higher than that obtained using the published Pore-C[57] technique (Fig. 1a).

To test whether repeated treatment can further reduce pore clogging, we carried out two and three rounds of simultaneous proteinase K digestion and reverse crosslinking (Fig. 1c and Supplementary Table 4). We successfully increased the sequencing output to an average of 128 Gbase and 144 Gbase, respectively (Supplementary Fig. 1d and Supplementary Table 4). However, the multiple rounds of proteinase K digestion and DNA purification are tedious and reduce the DNA recovery rate. To avoid these shortcomings, we first digested chromatin with proteinase K, then purified DNA and degraded peptides for another 40 min with pronase (Fig. 1c). Pronase is a mixture of nonspecific proteases from *Streptomyces griseus* that degrade both denatured and native proteins to nearly complete digestion into individual amino acids[59]. The purified DNA was sequenced, and an average of 128 Gbase data was generated per ONT PromethION cell run (Supplementary Fig. 1d and Supplementary Table 4). The number of multiway contact in HiPore-C and Pore-C reads is similar (Fig. 1h). Due to the increased sequencing throughput, pairwise contacts increased by 80% (Fig. 1d and Supplementary Table 4). Thus, we successfully developed two HiPore-C protocols that solved the pore-clogging problem (Fig. 1d and Supplementary Fig. 1c), further improved the

sequencing yield by about 80% compared to Pore-C (Fig. 1e) and virtual pairwise contact number, and reduced costs dramatically in both of the cell types that we tested (Fig. 1e–g, Supplementary Fig. 1d–f, Supplementary Table 2 and Supplementary Table 4).

We also developed the MapPore-C pipeline by integrating the third-generation sequencing programs NGMLR[60] and Minimap2[61] to map fragments in multiway contact reads to the reference genome (Supplementary Fig. 1g and Supplementary Table 5) and to generate virtual pairwise contacts (Supplementary Fig. 1h). We then evaluated the interexperimental variations during HiPore-C protocol development and showed that the datasets generated were highly correlated (Supplementary Fig. 1i, j). Thus, we combined them for further analyses.

Because of the low probability of interhomologous chromosome interactions (Supplementary Fig. 1k), theoretically, every molecule in an unamplified in situ HiPore-C library represents a unique array of multi-way-interacting DNA fragments from a single allele, thus allowing the exploration of single-allele topology in the cell population for genomic regions of interest. (Analyses below are carried out in GM12878 cells unless otherwise stated.)

### HiPore-C faithfully reproduces canonical 3D genome structures

To test whether HiPore-C can reproduce canonical 3D genome structures revealed by Hi-C, we first calculated Pearson's correlation coefficients and showed that the HiPore-C and Hi-C datasets[14] were highly correlated at both 500 kb and 50 kb resolutions in GM12878 cells (Fig. 2a, b and Supplementary Fig. 2a). Visual inspection of the HiPore-C pairwise contact map revealed typical chromatin structures including compartments A/B (Fig. 2c–e, Supplementary Fig. 2b, c), TADs (Fig. 2f, g, and Supplementary Fig. 2d), and chromatin loops (Fig. 2h, i, and Supplementary Fig. 2f, g) that were highly similar to those from Hi-C. Consistently, the HiPore-C and Hi-C pairwise contact maps were highly correlated at the levels of compartment eigenvector values ($r = 0.967$) (Fig. 2e) and TAD insulation scores (IS) ($r = 0.868$) (Fig. 2g). Pearson's correlation coefficients of the compartment eigenvector scores and TAD insulation scores together with the Hi-C dataset were calculated, and the correlations were high between pairs of replicates (Supplementary Fig. 2c, e). Together, these results prove that HiPore-C can faithfully capture typical 3D genome structures uncovered by conventional Hi-C.

### HiPore-C reveals interchromosomal chromatin clustering

We next asked whether HiPore-C can capture interchromosomal multiway contacts. Approximately 38% of reads contain fragments from nonhomologous chromosomes, the majority of which contain three or more fragments showing a positive correlation with inter-chromosomal interaction orders (Fig. 3a and Supplementary Fig. 3a), consistent with another study[62]. To characterize interchromosome interactions, we first separated genomic regions into telomeres, centromeres, and other genomic regions to plot the global inter-chromosomal contact matrix (Fig. 3b). Then, we calculated and determined the statistical significance of interchromosomal interactions for each pair of bins (1 Mb) (Supplementary Data 1). For telomeres, we detected a total of 109,941 pairwise contacts with telomere sequences at least at one end (Fig. 3c). Two thousand paired bins were significantly enriched with interchromosomal contacts, and only 41 of them had both ends located in telomeres (Fig. 3d, Supplementary Fig. 3b and Supplementary Data 2). For centromeres, we detected a total of 279,739 pairwise contacts with at least one end located in the centromere region (Fig. 3c). A total of 889 paired bins were significantly enriched with interchromosomal contacts, and 68 of them had both ends anchored in centromeres (Fig. 3e, Supplementary Fig. 3c and Supplementary Data 3). These results show that inter-telomere and inter-centromere contacts from nonhomologous chromosomes exist but only between a few chromosomes.

The majority of interchromosome pairwise contacts (3.69 million) occurred between genomic regions outside of telomeres and centromeres (Fig. 3c). We identified 34,654 interchromosomal bin pairs that were significantly enriched with pairwise contacts (Fig. 3c and Supplementary Data 1). We further separated bins involved in significant interchromosomal interactions into two clusters that formed hubs and those that did not (Fig. 3f and Supplementary Data 4).

Interestingly, cluster 1 interactions formed an inactive hub and bridged genomic regions mostly in small chromosomes (Fig. 3g and Supplementary Data 4). In contrast, cluster 2 interactions formed an active hub and connected both small and large chromosomes (Fig. 3h and Supplementary Data 4). Furthermore, gene density, enhancer density, and positive epigenetic modification levels were all higher in cluster 2 (Fig. 3i). As expected, the inactive cluster 1 hub mainly involved

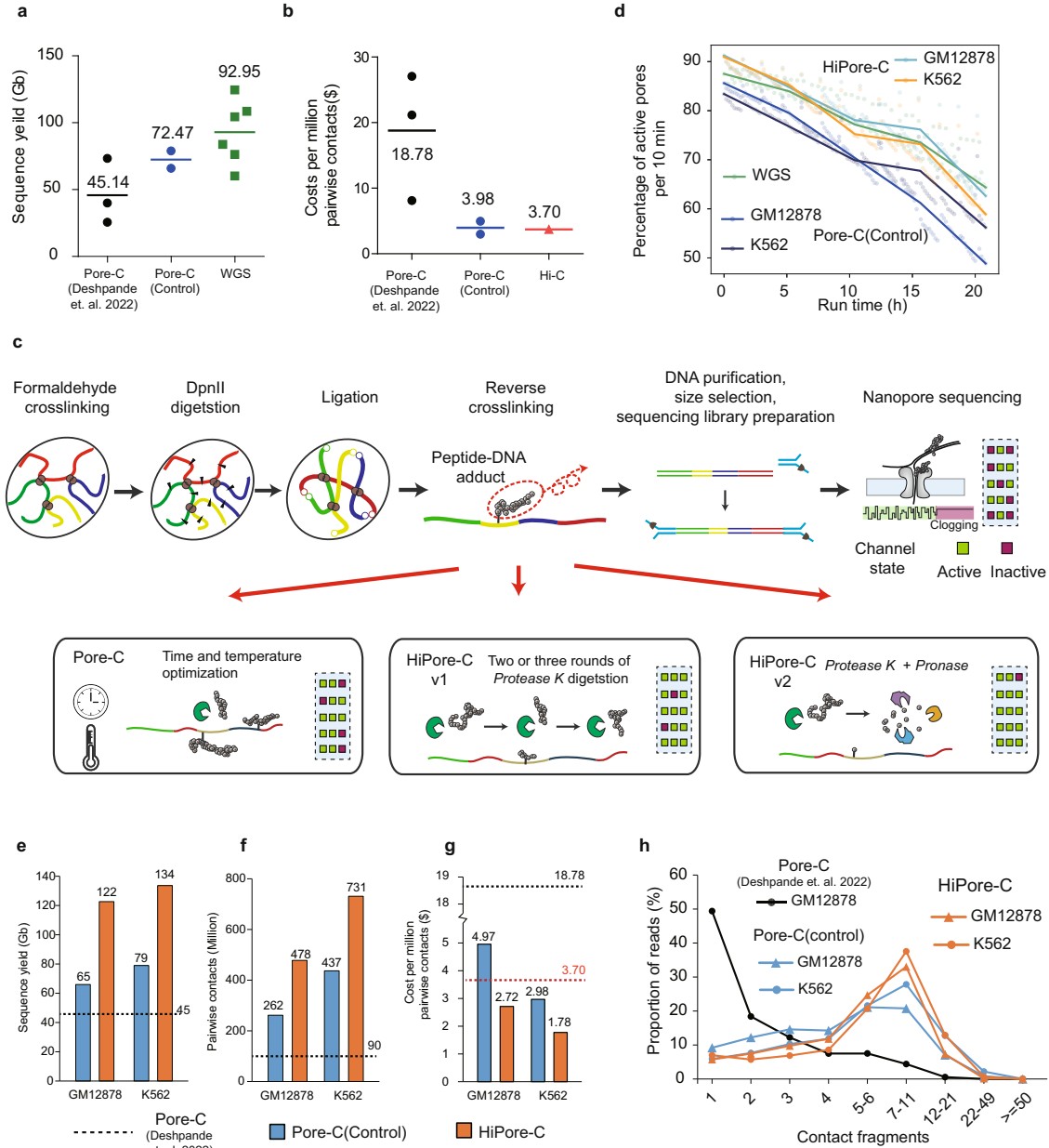

**Fig. 1 | Solving nanopore clogging increases the output of multiway contact sequencing. a** Comparison of the sequencing yield (Gb) between Pore-C and whole-genome sequencing (WGS) using ONT PromethION flow cells. Datasets of Pore-C (Deshpande et al., 2022)[57] and Pore-C (Control) were published and generated in this study, respectively (Pore-C[57], $n = 3$; Pore-C control, $n = 2$; WGS, $n = 6$). Lines indicate the mean values. Related to Supplementary Table 1. **b** Comparison of the costs per million pairwise contacts between Pore-C and Hi-C (Pore-C[57], $n = 3$; Pore-C control, $n = 2$). The cost of Hi-C is estimated based on the output of the Illumina Nova sequencing platform and the percentage of pairwise contacts that Hi-C can typically produce. Lines indicate the mean values. Related to Supplementary Table 2. **c** Schematic of the in situ HiPore-C protocols for generating higher-order chromatin interactions. Condition optimization for reverse crosslinking and the

effect of nanopore sequencing; Pore-C optimization, bottom left; HiPore-C v1, bottom middle, two or three rounds of reverse crosslinking and protease K digestion; HiPore-C v2, bottom right, reverse crosslinking plus *protease K* and *pronase* digestion. Flow cell sequencing channel clogging was compared. Green squares indicate active sequence channels; red squares indicate inactive channels. **d** Comparison of the decays of the percentage of active pores between WGS, Pore-C (Control), and HiPore-C. X-axis, sequencing time; y-axis, percentage of active pores per 10 min. **e-g** Comparison of the sequencing yield (Gb), the numbers of virtual pairwise contacts, and the costs between Pore-C and HiPore-C. Related to Supplementary Table 2. **h** Comparison of the distribution of numbers of ligated fragments in multiway contact long reads between Pore-C and HiPore-C.

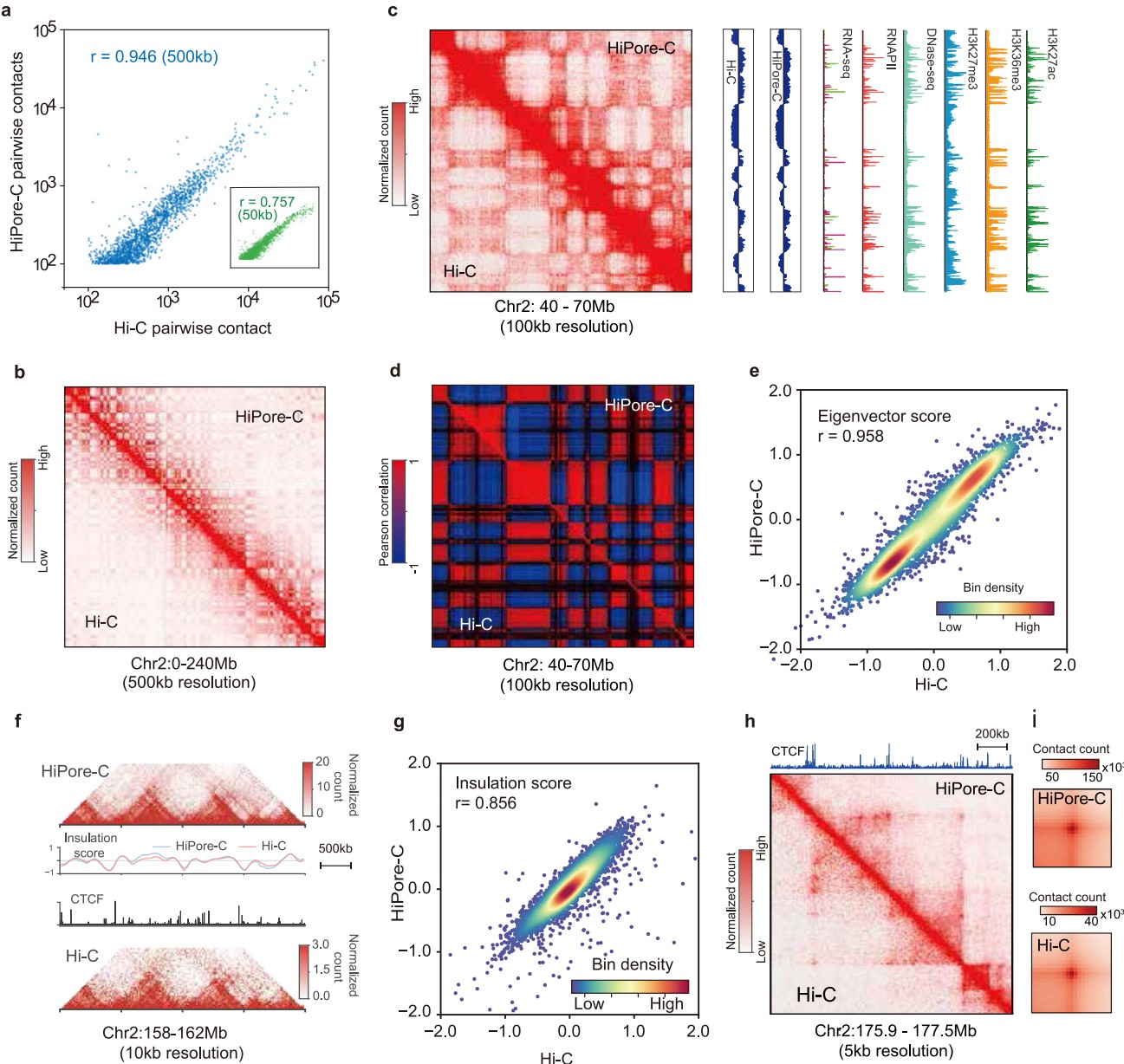

**Fig. 2 | HiPore-C faithfully reproduces canonical 3D genome structures.**
**a** Pearson's correlation of pairwise contacts between HiPore-C and Hi-C (Pairwise contact count > = 50; blue, 500 kb resolution, $n = 13,613,408$, $P = 0$; green, 50 kb resolution, $n = 12,498,125$, $P = 0$). **b** Comparison of contact maps between HiPore-C and Hi-C (upper right, HiPore-C; bottom left, Hi-C; 500 kb resolution). **c** An exemplary region showing compartment comparison between HiPore-C and Hi-C (100 kb resolution, tracks of eigenvector scores, RNA-seq, DNase-Seq, and ChIP-seq of RNA polymerase II (RNAPII), H3K27ac, H3K36me3, and H3K27me3 are shown on the right.). **d** An exemplary region showing a Pearson's correlation comparison between HiPore-C and Hi-C (100 kb resolution). **e** Correlation of eigenvector scores between HiPore-C and Hi-C (100 kb resolution bins, Pearson's correlation coefficient r = 0.958, $n = 27,935$, $P = 0$). **f** An exemplary region showing a TAD comparison between HiPore-C and Hi-C (10 kb resolution; insulation scores and CTCF track are shown in the middle). **g** Correlation of insulation scores between HiPore-C and Hi-C (50 kb resolution bins, Pearson's correlation coefficient r = 0.856, $n = 56,244$, $P = 0$). **h** An exemplary region showing a loop comparison between HiPore-C and Hi-C (10 kb resolution, CTCF track is shown at the top). **i** Comparison of the aggregate peaks between HiPore-C and Hi-C (10 kb resolution; peaks+/−100 kb).

compartment B segments. In contrast, the active cluster 2 hub mainly includes compartment A segments (Supplementary Fig. 3d). These results confirm the presence of two major inter-chromosomal hubs of different transcriptional activities[6]. In addition, we found that many tRNA genes were enriched in interchromosomal interactions, especially tRNA genes on chromosomes 1, 6, 14, 15, 16, 17, and 19 (Supplementary Fig. 3d, e, and Supplementary Table 6). These results suggest that interchromosomal interactions occur but generally at low rates for both constitutive heterochromatin of telomere, centromere, and nonrepetitive genomic regions.

## Multiway contacts span multiple compartments, TADs, and loops
Multiway chromatin interactions may span multiple 3D structural units of compartments, TADs, and loops, allowing direct measurement of the interaction frequency between individual 3D structural units across the whole genome[6,53,57,62,63]. To determine whether HiPore-C reads cover genomic distances long enough to cover multiple compartments, TADs, and loops, we first calculated genomic distances spanned by three types of fragment pairs (Fig. 4a). Overall, genomic distances covered by HiPore-C reads were positively correlated with

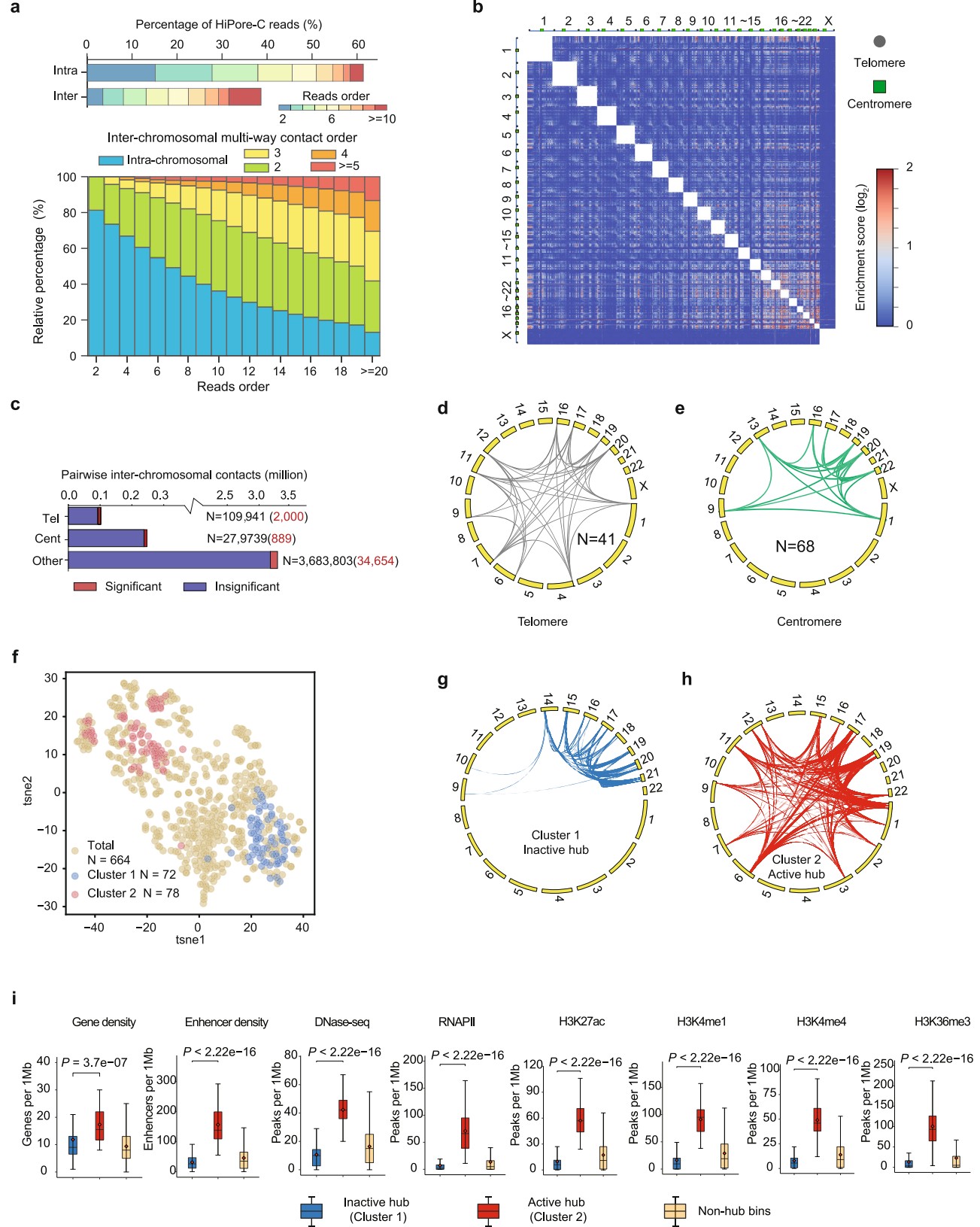

the number of fragments (Supplementary Fig. 4a–c) as reported in other studies[14,57]. The distances between nonadjacent fragments and between the most separated fragments in the multiway contacts were approximately 1 Mb in at least 50% of the HiPore-C reads (Fig. 4b–d). Although some compartments, TADs, and chromatin loops span genomic distances well over 1 Mb, their median sizes are 400 kb,

185 kb, and 274 kb, respectively (Fig. 4e). These results indicate that HiPore-C reads can be used to study the single-allele folding pattern over multiple 3D genomic structural units.

By comparing the heatmaps generated with adj- and non-adj-pairs of chromatin contacts (abbreviated as adj-pairs and non-adj-pairs), we showed that the overall chromatin interaction patterns were similar

**Fig. 3 | HiPore-C reveals interchromosomal chromatin clustering. a** Percentage of multiway intra-/inter-chromosome contact reads (top). The blue−red color bar represents the fragment number in reads. Percentage of reads of different numbers of fragments from different chromosomes (bottom). Different colors represent the number of chromosomes that a multiway contact read covers. **b** Global interchromosome interaction heatmap at 1 Mb resolution. The color bar represents the enrichment of interactions on a log scale. The chromosomes are labeled on top, green squares indicate centromeres, and gray dots indicate telomeres. **c** Interchromosome interaction counts, including contact bins located within centromere or telomere regions. The number of interactions is shown (Tel indicates telomeres interactions, $n = 109,941$; Cent indicates centromeres interactions, $n = 27,9739$; Other, interactions between genomic regions outside of the telomere and centromere sequences, $n = 3,683,803$). The number of significantly enriched interchromosome interactions was shown in red. Circos diagrams of significant interchromosome interactions between centromeres (**d**) or telomeres (**e**) at both anchor ends. **f** tSNE diagram of the significant interchromosome interaction regions. Two sets of interchromosome interaction regions are separated and labeled in red (active hub) and blue (inactive hub). Circos diagrams of two sets of interchromosome interactions: red, active hub (**g**), and blue, inactive hub (**h**). **i** Boxplots of gene density, enhancer density, DNase-seq, RNA Polymerase II, and histone modification (H3K4me3, H3K27ac, H3K36me3, H3K4me4, and H3K4me1) signals in inactive hub (blue, $n = 72$), active hub (red, $n = 78$), and control (yellow, $n = 2894$, genomic bins not in either of the active and inactive hubs) regions. The center line, median; red dot, mean; boxes, first and third quartiles; whiskers, 5th and 95th percentiles. Significance was calculated by the Kruskal−Wallis test, followed by Dunnett's $t$-test for active and inactive hubs. $P$ values are shown in boxplots. $P < 2.2 \times 10^{-16}$ in all of the Kruskal−Wallis tests.

and resembled Hi-C contact heatmap (stratum-adjusted correlation coefficients are 0.938, 0.808, and 0.844 for the heatmaps of adj-pairs and non-adj-pairs, adj-pair and Hi-C, and non-adj-pairs and Hi-C, respectively) (Supplementary Fig. 5a). We further compared the structures of compartments, TADs, and loops. In all cases, structural patterns generated using adj-pairs, non-adj-pairs, and Hi-C datasets showed strong correlations (Pearson's correlation coefficients are 0.919, 0.942, and 0.982 for eigenvector scores, and 0.677, 0.706, and 0.902 for insulation scores between the non-adj-pairs and Hi-C, adj-pairs, and Hi-C, and adj-pairs and non-adj-pairs, respectively) (Supplementary Fig. 5b, c). In addition, we could identify the same loops using adj- and non-adj-pairs (Supplementary Fig. 5d−e). The fact that no apparent differences were observed suggests that non-adj pairwise contacts are not fundamentally different from the classical direct adj-ligations in single reads. Thus, we conclude that the non-adj-ligations can be considered chromatin "contact" at least at the resolutions we analyzed the data.

Although overall chromatin interaction patterns are similar between chromatin interaction matrices generated from adj- and non-adj-chromatin interaction pairs, we did find that adj-pairs were more enriched within the same structural unit while non-adj-pairs were more enriched in reads spanning multiple structural units (for adj- and non-adj-pairs: inter-chromosomal enrichment scores are 0.45 and 1.17; inter-compartment enrichment scores are 0.599 and 1.132 (A-A), and 0.775 and 1.073 (B-B), respectively; inter-TAD enrichment scores are 0.750 and 1.081) (Supplementary Fig. 5f−h). Overall, non-adj contacts are more enriched in reads covering multiple structural units than adj- and conventional Hi-C pairwise contacts. More importantly, the fragments seem to be arranged orderly in the sequenced long-reads supporting a previously proposed conjecture that the linked segments are not randomly distributed but comply with the chromatin extension paths like C-walks, and the fragment arrangement order could have important spatial and biological implications that require further investigation[63].

We first examined two previously identified adjacent loops to measure the loop anchor interaction frequency. Out of a total of 10,113 HiPore-C reads containing fragments of at least one anchor (A, B, or C), most reads (9586, 94.79%) contained only one of the three anchor fragments (Fig. 4f). Only 4.95% (501/10113) of reads contained two anchors (A-B, A-C, and B-C), and even fewer (0.26%, 26/10113) reads contained three anchors (Fig. 4f). Although the formation of one loop requires two anchors, the two loop anchors do not necessarily coexist in the same read in our HiPore-C analysis because loops are identified based on pairwise interactions derived from all contacts in HiPore-C reads. We found that 50.5% of HiPore-C reads contained one loop anchor, with 37.0% of reads containing an anchor for only one loop and 13.5% of reads containing an anchor for multiple loops that shared the same anchor (Supplementary Fig. 4d−g). Reads containing both anchors of a loop accounted for 3.3% of total reads, including 0.27% of total reads that contained anchors for multi-loops. That 53.6% of reads

contain certain anchor sequences for loops suggests that looping is a general principle of chromosome folding. At the same time, the low percentage of reads containing both anchors of a loop or anchors of multiple loops suggests that loop formation could be very dynamic, consistent with the observation in Fig. 4f. The low coexistence of anchor fragments in multiway interaction reads is consistent with a recent live microscopic observation showing that even strong intra-chromosomal interactions occur in only ~3% of cells[64]. Using multiway contacts, we also identified higher-order interactions of consecutive loops[6]. Nevertheless, these results show that HiPore-C multiway interaction reads can be used to calculate the interaction probability between any two genomic loci across the whole genome in a population of cells, a task that has only been feasible now.

TADs contain self-associating chromatin restricted to a discrete genomic region. However, long-range chromatin interactions that anchor in one TAD and reach out into genomic regions in other TADs must occur to establish 3D genome structures of compartments and chromatin loops. To answer this puzzling question, we extracted 49,065 multiway HiPore-C reads that each contained at least two fragments located in a genomic region on chromosome 2 (98.58-99.37 Mb) that covered four TADs (Fig. 4g). Interestingly, only 9.17% (4500/49,065) of HiPore-C reads contained fragments exclusively within only one of the four TADs (average 2.3%, 1125/49,065 per TAD). Most multiway interaction reads (69.08%, 33,892/49,065) contained at least one fragment in a TAD outside of this analyzed genomic region. Additionally, 21% (10,673/49,065) of HiPore-C reads contained fragments in two, three, and all 4 TADs within this genomic region. At the genome-wide scale, approximately 54% of reads span two or more TADs (Supplementary Fig. 4h). The number of fragments in a read positively correlates with the number of TADs being covered, in agreement with the results in Fig. 4g. These results suggest that single alleles may fold dynamically into different forms of "loop-string-loop" structures in which a read contains two loops established by two pairs of fragments that are far-separated in the linear genomic distance. Interestingly, most of these structures represent interactions between fragments from TADs separated by more than one TAD. Like a previous study[62], our HiPore-C data also support that intra-TAD interactions synergize with inter-TAD long-distance interactions to form a higher-order 3D genomic structure.

We further asked whether single-allele chromatin interactions are mostly confined within one compartment or span both types of compartments. To address this question, we chose a genomic region (53.83−60.53 Mb) on chromosome 14, extracted HiPore-C reads with at least two fragments falling within this region, and clustered HiPore-C reads based on their fragment distribution in the A and B compartments (Fig. 4h). A total of 55.21% (155,987/282,523) of HiPore-C reads contained fragments located in only compartment A or compartment B. Consistent with a previous study[62], a higher percentage (56.74%, 40,684/71,707) of compartment B reads contained fragments located in multiple B compartments than the percentage of compartment A

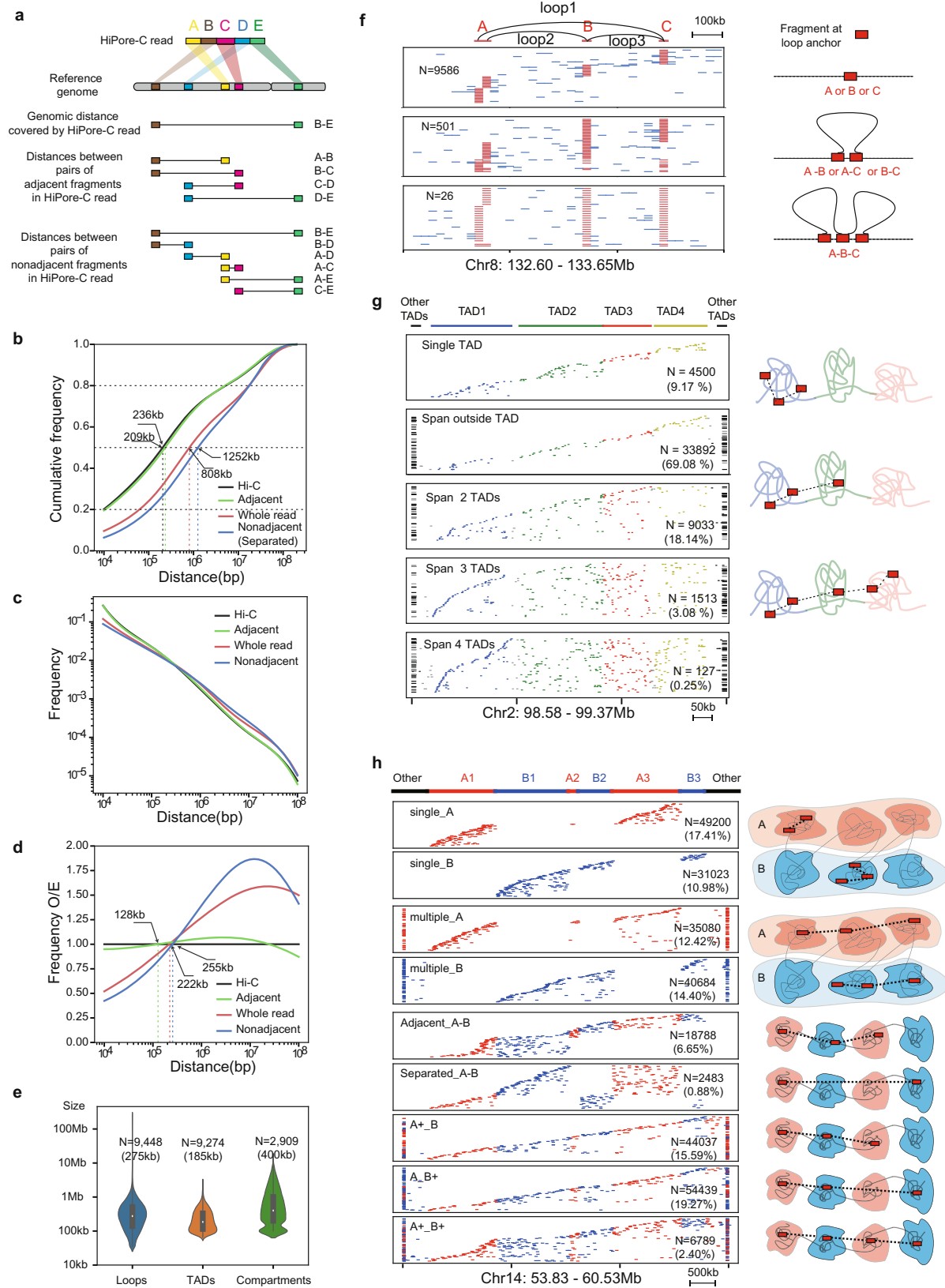

reads (41.62%, 35080/84280) that contained fragments in multiple A compartments suggesting that repressive chromatin may associate more easily than active chromatin. Less than 50% (44.79%, 126,536) of HiPore-C reads contained fragments in both the A and B compartments. In fact, among these reads, 34.8% (44,037/126,536) and 43.02% (54,439/126,536) showed the pattern "multi A-one B" or "multi B-one

A", respectively. Reads with fragments located in "multi A-multi B" compartments were rare (5.34%, 6789/126,536). Reads containing fragments in adjacent or separated A and B compartments were also infrequent (14.85%, 18788/126,536; 1.96%, 2483). The genome-wide analysis produced similar results (Supplementary Fig. 4f–i). It confirmed that interactions spanning the same-type compartment (A-A

**Fig. 4 | Multiway contact reads span multiple compartments, TADs, and loops.**
**a** Schematic diagram showing how three types of contact distances were calculated: 1. the longest genomic distance covered by fragments in a read; 2. genomic distances between pairs of adjacent fragments in a read; and 3. genomic distances between pairs of nonadjacent fragments in a read. **b** Cumulative frequency of different types of paired fragments against genomic distance for HiPore-C and Hi-C data. **c**, Decaying curves of different types of paired fragments in HiPore-C and Hi-C data. **d** Frequencies of different types of paired fragments normalized against those of Hi-C data over continuous genomic distance. **e** Violin plots showing the size distribution of chromatin loops, TADs, and compartments. Total of 9448 loops, 9274 TADs and 2909 compartments were identified[14]. The average size of each structural unit was shown. The center dot, median; boxes, first and third quartiles; whiskers, 5th and 95th percentiles. **f** An exemplary region showing multiway contact

reads spanning three loop anchors (bin size is 25 kb). The top panel shows the loop interaction arcs. Reads covering different numbers of loop anchors are shown separately in each panel. Read numbers are shown in each panel. **g** An exemplary region showing multiway contact reads spanning multiple TADs (bin size is 10 kb). Colored fragments correspond to the TADs in which they are located. Fragments outside the analyzed region (Chr2: 98.58-99.37 Mb) are marked in black. Reads spanning different numbers of TADs, and the corresponding number of reads are shown separately in each panel. **h** An exemplary region showing multiway contact reads spanning multiple compartments (bin size is 10 kb). Colored fragments and reads correspond to the type of compartment in which they are located. Fragments outside the analyzed region (Chr14: 53.83-60.53 Mb) are marked in black. Reads spanning the compartments, and the corresponding numbers of reads are separately grouped and shown in each panel.

and B-B) were more frequent than interactions spanning both A and B compartments[62,63]. These results confirm that multiway interactions of a single allele are not random and preferentially confined to a specific type of compartment.

These analyses could be successfully carried out because the greater the number of fragments in a HiPore-C read, the more loop anchors, TADs, and compartments it may span (Supplementary Fig. 4d–f). However, the number of HiPore-C reads decreases as the numbers of loop anchors, TADs, and compartments that can be covered by single-allele reads increase (Supplementary Fig. 4g–i), highlighting the importance of producing high-order fragment interactions within each read.

### Diversity and cell type-specificity of single-allele topology clusters underlie the formation of TADs

TADs are highly similar in different cell types and even in different organisms[13,65,66]. However, microscopic imaging analyses indicate that the TAD border can be promiscuous, suggesting a lack of homogeneity in chromatin folding in single cells[67]. We wondered whether high-order reads might reflect finer structures inside TADs. First, we confirmed that hierarchical clustering could successfully separate high-order HiPore-C reads into single TADs (Supplementary Fig. 6a, b). Next, we chose a TAD (70.18-70.42 Mb) on chromosome 11 that is nearly identical in GM12878 and K562 cells (Fig. 5a, b). HiPore-C reads were clustered into three groups, with most fragments preferentially confined within a sub-TAD range. These three clusters of reads correspond to pairwise contact matrices that differ between the two cell types (Fig. 5c, d). In both cell lines, cluster 2 was between pairs of tandem CTCF sites with the lowest number of reads (25%, 454/1813 in GM12878; 28.1%, 667/2374 in K562). Cluster 3 was shorter in GM12878, with 29% (523/1813) of reads than in K562 (34.5%, 818/2374). Cluster 1 was more prominent in GM12878, with 46% (836/1813) reads than in K562 (37.4%, 889/2374). Interestingly, fragments containing CTCF motifs and pairwise interactions between them were many-fold higher in K562 cells than in GM12878 cells. This difference correlates with the varied gene transcription in the region of cluster 1 reads. Thus, we show that a single allele may adopt several preferred topologies in a cell type-specific manner in conserved and highly similar TADs.

In addition, we examined the human *Fbn2* TAD (similar to the mouse *Fbn2* TAD[64]). We again revealed differences in single-allele topology preference despite silent gene expression in this TAD in both GM12878 and K562 cells (Supplementary Fig. 6c–g). Thus, we conclude that fragments in single alleles tend to cluster in discrete regions. Within each cluster, the single-allele topology can be highly diverse. However, suppose one cluster contains enough fragments generally clustered in neighboring or even more distant regions; in that case, these clusters will not be identified as separate TADs in the pairwise contact matrices. Otherwise, these clusters can be identified as separate TADs.

To further test this hypothesis, we dissected a hierarchical TAD (121.34-121.81 Mb) on chromosome 2 in GM12878 (Fig. 5e)[14].

Interestingly, HiPore-C reads were clustered into three groups instead of two corresponding to the two visually identifiable sub-TADs. The contact matrices of cluster C2 and C3 reads showed numerous outreaching interactions over cluster C1 in the middle (Fig. 5f). Consistently, genomic distances covered by HiPore-C reads and pairwise fragments in the C2 and C3 clusters spanned much longer distances at higher frequencies than C1 reads (Fig. 5g). These results show that single alleles in the sub-TADs of a hierarchical TAD form a curved dumb bell-like structure in which clustered multiway contacts located at the two ends of a TAD frequently colocalized in the same reads (Fig. 5f) implying they could interact more frequently than with the sequences separated them in the middle of a TAD (Fig. 5g), forming a bent dumb-bell whose two ends meet. In addition, we also noticed that CTCF pairwise interactions in single HiPore-C reads varied dramatically in GM12878 and K562 cells (Fig. 5a and Supplementary Fig. 6c, d). Surprisingly, intra-TAD clusters of single-allele topologies do not correlate with convergent CTCF binding, suggesting that other mechanisms dictate the topology choices within restricted regions in a TAD. Nevertheless, these results are consistent with our model that relations between clusters of single-allele topologies underlie TAD partitioning.

### HiPore-C reveals a cell type-specific enhancer hub at the β-globin locus

To test whether high-order HiPore-C reads may capture functionally relevant 3D structures, we compared the human β-globin locus in K562 and GM12878 cells. Human embryonic ε-, fetal Gγ- and Aγ-globin genes were expressed in K562 cells but not in GM12878 cells, and pairwise contact matrices of the β-globin locus showed no obvious differences[68] (Fig. 6a, b, Supplementary Fig. 7a, b). HiPore-C reads in this region were clustered into two groups. Cluster 1 (C1) contains hypersensitive sites 5-3 (HS5-3), skips over cluster 2 (C2), and covers adult δ- and β-globin genes and 3′HS1. C2 (32.3%, 985/3052) covers a genomic region between the downstream region of HS3 and the upstream region of the silent δ-globin gene in K562 cells (Fig. 6a and Supplementary Fig. 7c). In GM12878, the majority of reads were in cluster 2 (74.3%, 2218/2985), covering the sequences from upstream of 5′HS5 to downstream of the β-globin gene, with cluster 1 covering the rest of the β-globin locus, including 3′HS1 (Fig. 6b and Supplementary Fig. 7d). Interestingly, C2 in K562 cells contains HS2 and HS1 but not HS3-HS5, suggesting that HS2 and HS1 in the LCR physically interact with and enhance embryonic and fetal globin gene expression (Fig. 6c and Supplementary Fig. 7e). Interactions among ε- and Gγ-/Aγ-globin genes, HS2, HS1, and the region upstream of the ε-globin gene in K562 were much less frequent in GM12878 (Fig. 6d and Supplementary Fig. 7f). In addition, three-way interaction analysis confirmed the coexistence of the HS2-HS1, ε-globin gene, and Gγ-/Aγ-globin genes in C2 reads, especially in K562 cells (Fig. 6e, f). Consistent with several multi-contact studies of the β-globin locus[43,55], globin gene promoters and enhancers can interact simultaneously to form an enhancer hub. We also found that the HS5-HS3, HS2-HS1, and ε-globin genes coexist but at a lower rate (Fig. 6g, h), suggesting that HS5-HS3 are less

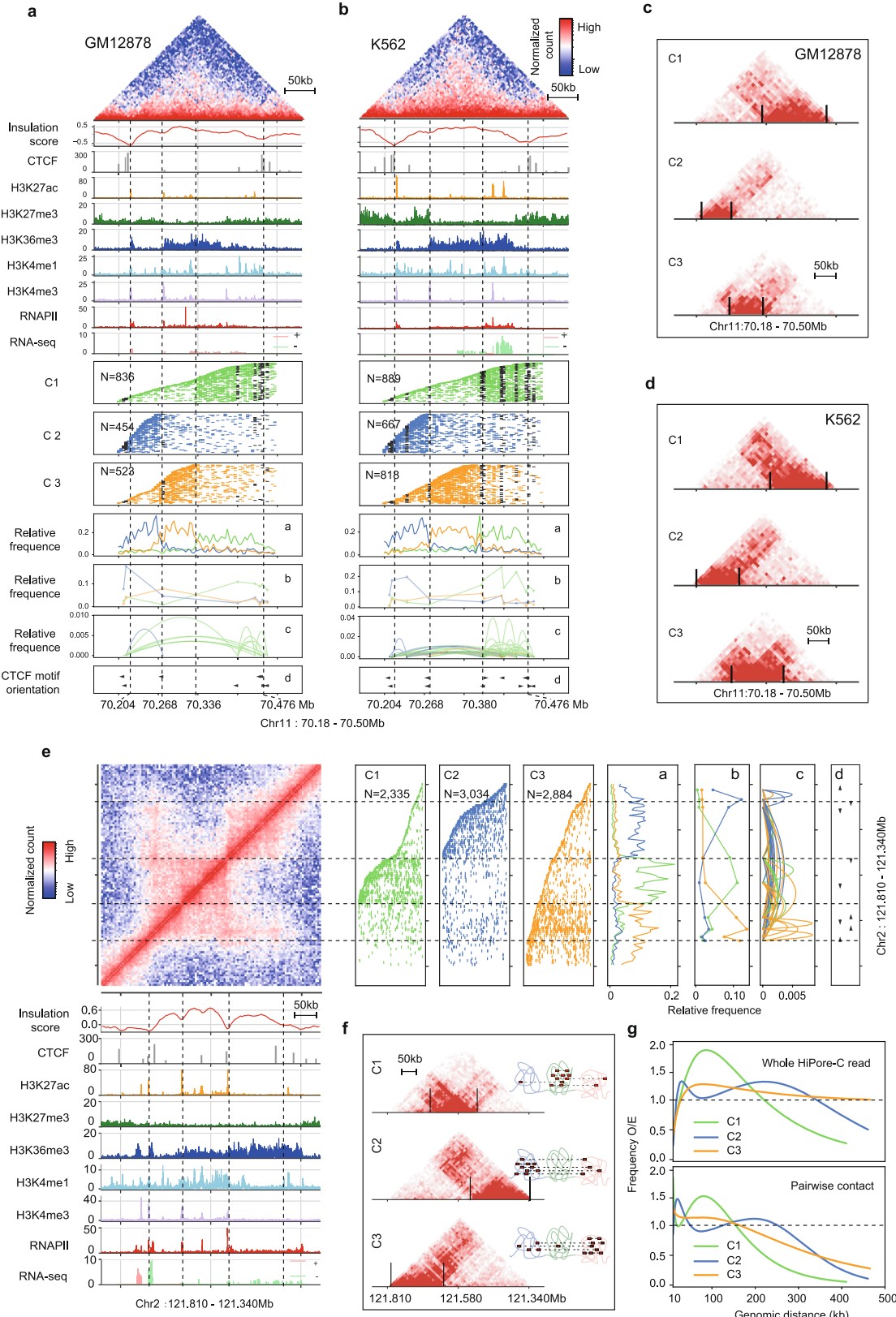

involved in the enhancer hub that activates ε-, Gγ- and Aγ-globin gene expression. The silent adult δ- and β-globin genes and 3′HS1 showed a much weaker interaction in C2 in both K562 and GM12878 cells (Supplementary Figs. 6i, 7j). The fact that only 32.3% of alleles adopt a C2 topology in K562 cells suggests that chromatin interactions are dynamic and short-lived, consistent with the microscopic observation

that even strong interactions between CTCF sites exist in only 3% of cells and last for only 20-30 min[64]. We conducted multiple promoters and enhancer interaction analyses as described[27]. Our results also revealed a low proportion of multiway promoter and multiway enhancer interactions (Supplementary Fig. 8a−c and Supplementary Data 5, 6). Consistent results were also obtained in promoter and

**Fig. 5 | Diversity and cell type-specificity of single-allele topology clusters underlie the formation of TADs.** Clustering of HiPore-C reads covering an exemplary TAD in GM12878 (**a**) and K562 (**b**) at 5 kb resolution (bin). The top panels are the pairwise contact heatmaps and additional tracks, as indicated. Middle panels are hierarchical clusters of multiway contact reads: C1, green; C2, blue; C3, orange. Reads are arranged according to the genomic coordinates of the 5′ fragments in reads. Bins containing CTCF peaks (ENCODE data) are shown in black. The relative frequencies of bins (**a**) and bins containing CTCF (**b**) are shown in the line plot panel. The relative frequency of pairwise contact between bins containing CTCF peaks (**c**) is shown in the arc-line panel. The orientation of CTCF motifs (**d**) that are covered by fragments in HiPore-C reads is shown in the bottom panel. **c, d.** Heatmaps of the three HiPore-C read clusters in panels **a** and **b. e**, Clustering of HiPore-C reads covering a hierarchical TAD in GM12878 (5% of reads from each cluster were randomly selected and visualized in panels C1-C3). **f**, Heatmaps of the three HiPore-C read clusters in panel **e. g.** The normalized observed/expected contact frequency against genomic distance for three clusters of reads in the hierarchical TAD shown in panel **e**. The top panel shows the genomic distances covered by whole HiPore-C reads. The bottom panel shows the genomic distances covered by virtual pairwise contacts in HiPore-C reads.

enhancer multiple interaction analysis of two well-studied gene families of the Histone gene 1, 2, 3 (*HIST1*) and the human leukocyte antigen (*HLA*) gene loci (Supplementary Figs. 9 and 10). Altogether, these results demonstrate that HiPore-C can reveal functionally relevant structural details and heterogeneity in single-allele topology at an unprecedented resolution.

### HiPore-C captures DNA methylation and chromatin topology simultaneously

ONT sequencing can detect DNA methylation directly. To test whether HiPore-C can capture DNA methylation faithfully, we processed HiPore-C ONT sequencing signals and obtained highly reproducible methylated CpG profiles (Supplementary Fig. 11a, b) that were highly consistent with DNA methylation profiled by whole-genome bisulfite conversion sequencing (WGBS) (ENCODE ENCFF067JYV) (Fig. 7a). At both high and low methylation levels, the majority of CpG methylation sites were captured by HiPore-C (Fig. 7b) and highly correlated with the WGBS data (Pearson's correlation, r = 0.8038) (Fig. 7c). These results prove that HiPore-C can faithfully capture DNA methylation just as it can faithfully capture 3D genome structures.

DNA methylation is prevalent in the human genome and enriched in various functional genomic regions that may fold into distinct 3D structures. We first examined and showed a positive correlation of DNA methylation at chromatin loop anchors (Fig. 7d, e, Pearson's correlation, r = 0.119). We further separated loops into three groups with or without the CTCF motif. Anchors with CTCF motifs at both anchors showed the lowest DNA methylation levels, possibly because CTCF binding can be blocked by DNA methylation in its motif, and anchors without CTCF motifs showed the highest DNA methylation level (Fig. 7f and Supplementary Figs. 11c–e). The correlation of DNA methylation levels at two anchors was also the highest in non-CTCF loops and the lowest in loops with CTCF motifs at both anchors (Fig. 7g). Together with DNA methylation, DNase I hypersensitivity, H3K27ac, and RNA expression were all positively correlated at loop anchors (Supplementary Fig. 11f–k), suggesting that looping facilitates long-range co-modification of chromatin.

Compartment A contains a higher density of genes than compartment B, and DNA methylation is enriched in the mammalian gene body, suggesting that compartments A and B can be determined based on DNA methylation level. To test this hypothesis, we first compared the methylation levels in compartments A and B[14]. As expected, the DNA methylation level was significantly higher in compartment A (Fig. 7h). We then used DNA methylation level to determine the compartment types and showed that more than 93% of the compartments could be reproduced (Fig. 7i, j). A zoomed-in view of a genomic region shows DNA methylation enriched in the gene body and devoid at the promoter with H3K27ac (Fig. 7k, l), indicating the association between DNA methylation and the gene body. These results prove that HiPore-C sequencing can accurately determine compartment types by simultaneously measuring DNA methylation levels.

### Discussion

Here, we described HiPore-C, an assay that simultaneously captures multiway higher-order chromatin interactions and DNA methylation in populations of cells in one experiment. HiPore-C provides more virtual pairwise chromatin interactions than traditional Hi-C and Pore-C for the same cost through a much simpler procedure.

HiPore-C captures multiway chromatin interactions. Theoretically, any two multiway long reads covering a specific genome region can be estimated to be allele-specific or not if the cell population is large enough, especially if there is an overlap of sequences between the two reads, allowing the study of single-allele topology for any designated genomic region. Because of this remarkable feature, HiPore-C allows the exploration of genome folding principles at an unprecedented resolution and helps address a few long-standing questions.

HiPore-C shows that a typical chromatin structure TAD contains multiple clusters of distinct multiway chromatin interactions. Each cluster of interactions forms a partial pattern of a TAD. Only after the aggregation of all the patterns can a typical TAD be observed. Interestingly, a sub-TAD in a hierarchical TAD can present a bent dumbbell-shaped structure represented by one cluster of single alleles. Another cluster of single alleles represents another local sub-TAD in the middle. This unexpected discovery implies that each allele's dynamic folding can be more complex than previously thought.

The capability of capturing the single-allele topology of HiPore-C data also allows an in-depth investigation of the 3D genome structure's role in gene regulation. Using the human β-globin locus as a model, we reveal the heterogeneity of local allele-specific chromatin interactions and show that only a subset of interactions may support ε-, ᴳγ-, and ᴬγ-globin gene expression by bringing enhancers in the LCR to these target genes. For many alleles, the 3D structures suggest a lack of communication between enhancers and target genes. However, it is difficult to distinguish at this stage whether the transcription-supportive and inactive structures can dynamically transit between each other or remain unchanged in an allele-specific manner and whether these structures reflect the states of alleles in cells at different cell cycle phases. Nevertheless, our HiPore-C results greatly improve our understanding of the complexity of the 3D local chromatin structure and its relationship with transcriptional regulation.

HiPore-C is a powerful tool for higher-order genome structure mapping in 3D space. In addition to its current application, HiPore-C can be modified in a few ways. For example, single-cell RNA-seq and single-cell HiPore-C can be combined to reveal whether allele-specific chromatin structures correlate with variations in RNA expression in single cells. In addition, HiPore-C can be modified to generate combinatorial maps of DNA accessibility, RNA loops, histone variants/modifications, or transcription factors with high-order 3D structures. These potential applications will empower the exploration of the elusive mechanisms of 3D structure establishment and the relationship between spatial genome organization and gene regulation in the nucleus during development and differentiation.

### Methods
#### Cell culture
Human B lymphocyte GM12878 cells (Coriell Institute) and erythroleukemia K562 cells were incubated in 1× RPMI 1640 media supplemented with 15% (GM12878) or 10% (K562) fetal bovine serum at 37 °C with 5% $CO_2$.

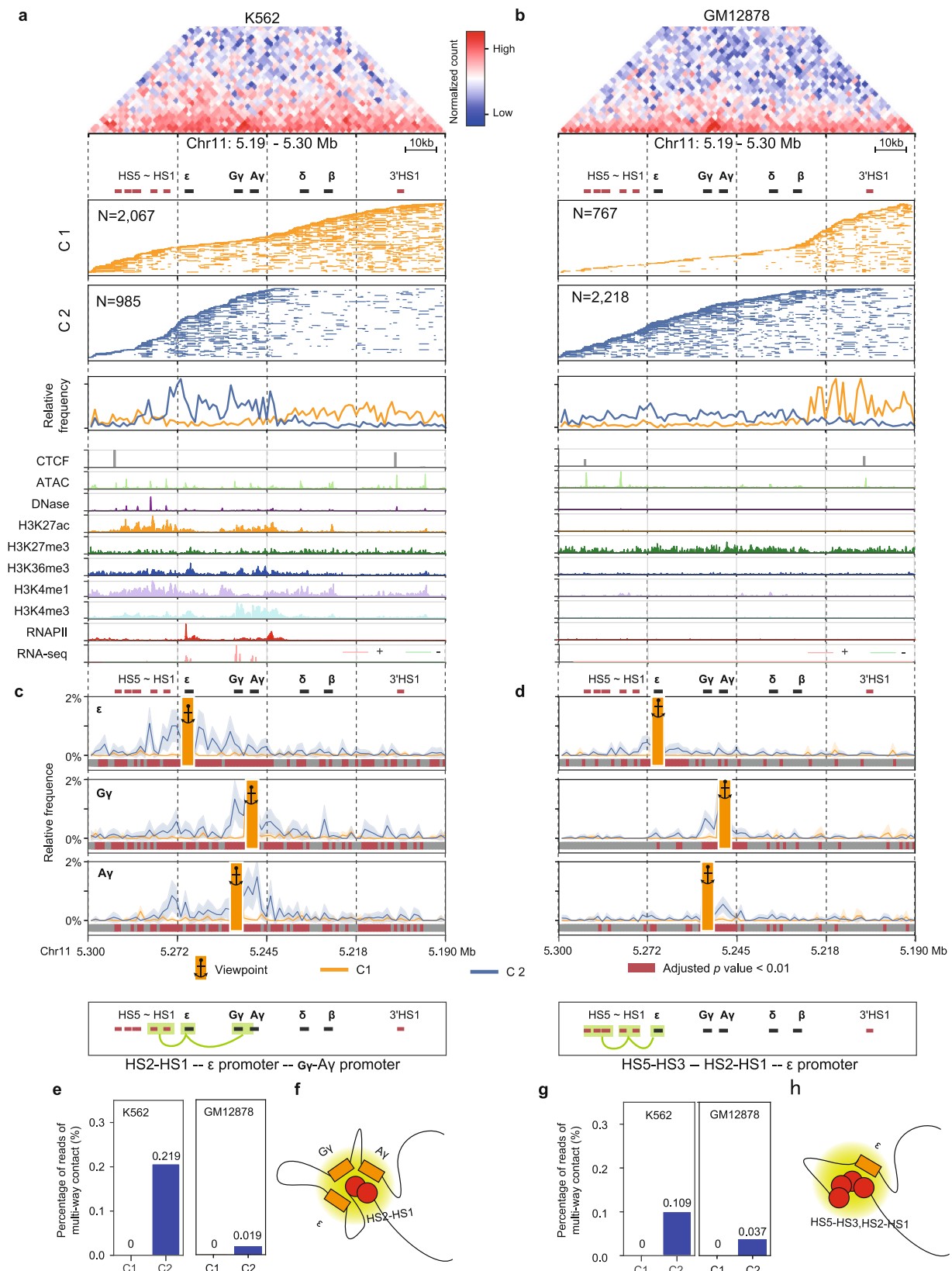

## Cell crosslinking

Fifteen million GM12878 or K562 cells were spun down and resuspended in 10 ml of fresh medium. Cells were fixed by adding 278 µL of 37% formaldehyde and incubated for 10 minutes at room temperature (RT). The reaction was stopped by adding 894 µL of 2.5 M glycines. The

cell suspension was incubated for five minutes at RT, followed by 10 minutes on ice. Fixed cells were pelleted by centrifugation at 1000 × g for 5 minutes at 4 °C and then gently washed twice with 5 ml of ice-cold 1× PBS. The cell pellet was stored at −80 °C until further processing.

**Fig. 6 | HiPore-C reveals a cell type-specific enhancer hub at the β-globin locus.** Clustering of HiPore-C reads covering the human β-globin locus in K562 (**a**) and GM12878 (**b**) cells (The HiPore-C reads were binned at 1 kb resolution. Only reads containing three or more fragments in the region of Chr11:5.19-5.30 Mb were used to perform clustering. Fifty percent of the reads from each cluster were randomly selected and visualized in multi-contact read panels). Multiway contact analysis anchored at human ε- and Gγ-/Aγ-globin gene promoters in K562 (**c**) and GM12878 (**d**) cell lines. Viewpoints are shown as anchors. Reads from each cluster were randomly sampled 100 times to generate subsample sets. The relative appearance frequency of reads with viewpoints was calculated. Lines with shading represent the mean±sd of the bin relative appearance frequency in the subsample sets. The statistical significance of the relative appearance frequency of bins was calculated by comparing the two clusters using a two-sided Welch's test with Bonferroni correction and is depicted with gray and dark red bars (gray, non-significant, adjusted $P = 0.01$; red, significant, adjusted $P < 0.01$). **e** Multiway contacts at the human β-globin gene locus formed by simultaneous interactions of human ε- and Gγ-/Aγ-globin genes and two hypersensitive enhancer sites (HS2 and HS1). **f** A graphic showing the multiway contacts formed by simultaneous interactions of three human ε- and Gγ-/Aγ-globin genes and two hypersensitive enhancer sites (HS2 and HS1). **g** Multiway contacts at the human β-globin gene locus formed by simultaneous interactions of the human ε-globin gene and hypersensitive sites (HS2, HS1, and HS5-HS3) in the locus control region. **h** A graphic showing the multiway contacts in **g**.

## Chromatin digestion and ligation

Up to three million crosslinked cells were resuspended in 1000 μL of ice-cold cell lysis buffer (10 mM Tris-HCl pH 7.5, 10 mM NaCl, 0.2% NP-40, 1× Roche protease inhibitors) and rotated at 4 °C for 30 min. Nuclei were pelleted at 4 °C for 5 min at 1000 × $g$, and the supernatant was discarded. Pelleted nuclei were washed once with 500 μL of ice-cold cell lysis buffer. The supernatant was removed, and the nuclear pellet was resuspended in 50 μL of 0.5% SDS and incubated at 62 °C for 10 min. Then, 145 μL of water and 50 μL of 10% Triton X-100 were added, and the samples were rotated at 37 °C for 15 min to quench SDS. Then, 25 μL of 10x NEB Buffer 3.1 and 10 μL of 10 U/μL DpnII restriction enzyme (NEB, R0543T) were added, and the sample was rotated at 37 °C for 4 h. DpnII was then heat-inactivated at 62 °C for 20 min. Then, the reactions were rotated at 4 °C for 5 min. A total of 750 μL of ligation master mix was added: 100 μL of 10× NEB T4 DNA ligase buffer with 10 mM ATP (NEB, B0202), 75 μL of 10% Triton X-100, 3 μL of 50 mg/mL BSA (Thermo Fisher, AM2616), 10 μL of 400 U/μL T4 DNA Ligase (NEB, M0202), and 562 μL of water. The reactions were rotated at 16 °C for 4 h and then allowed to proceed for an additional 1 h at RT.

## DNA purification procedure optimization

We added 45 μl of 10% SDS and 55 μl of 20 mg/ml proteinase K to reverse crosslinking of the ligated chromatin. Samples were incubated at 63 °C for at least 4 hours (overnight recommended). Then, we added 65 μl of 5 M NaCl and incubated the samples at 68 °C for at least 2 hours. Next, samples were extracted with 500 μl of phenol:chloroform: isoamyl alcohol (25:24:1). After centrifugation at top speed, the aqueous phase was separated using a 2 ml MaXtract high-density tube. Then, 1 μl of GlycoBlue, 100 μL of 3 M sodium acetate (pH 5.2), and 850 μL of isopropanol were added to the aqueous solution. The mixture was incubated at −80 °C for 1 hour. We centrifuged the mixture at maximum speed for 30 minutes at 4 °C, removed the supernatant, and washed the pellet twice with ice-cold 75% ethyl alcohol before dissolving the dried pellet with 170 μL of Buffer EB. The above is the optimized Pore-C experimental protocol.

For version 1 HiPore-C, we repeated digestion by adding 20 μL of 10% SDS and 10 μL of 20 mg/ml proteinase K to 170 μL of DNA solution. The mixture was incubated at 63 °C for 1 hour to digest the remaining associated protein and purified as in the first round. Proteinase digestion and reverse crosslinking can be repeated for another round. The final library DNA was dissolved in 30 μL of Buffer EB.

For version 2 HiPore-C, we digested samples for an additional round with pronase and then purified library DNA as described in the Pore-C and HiPore-C version 1 protocols. The final library DNA was dissolved in 30 μL of Buffer EB.

## Nanopore sequencing library preparation and ONT single-molecule sequencing

3-4 ug of purified DNA per sample was used as input material for ONT sequencing library preparation. DNA was size selected (>3 kb) using the PippinHT system (Sage Science, USA). DNA ends were repaired with dA addition, and the A-ligation reaction was conducted with the NEBNext Ultra II End Repair/dA-tailing Kit (Cat# E7546). The adapter in SQK-LSK109 (Oxford Nanopore Technologies, UK) was used for further ligation, and the DNA library was measured on a Qubit 4.0 fluorometer (Invitrogen, USA). Approximately 700 ng of library DNA was sequenced on the ONT PromethION (or MinION) platform at the Genome Center of Grandomics (Wuhan, China). And we carried Pore-C experiment described by Deshpande et al[57]. on GM12878 and K562 cell lines and sequenced these libraries on the PromethION platform for comparison with the HiPore-C.

## Nanopore sequence base-calling and methylation calling

Nanopore sequencing raw signals were converted to DNA sequences using the high-accuracy model "dna_r9.4.1_450bps_hac_prom.cfg" of Guppy v4.5.3 software (Oxford Nanopore Technologies) and reads with quality scores less than 7 were discarded. Sequencing statistical analysis was conducted using NanoPlot[69]. 5mC methylation sites were called using Megalodon (Oxford Nanopore Technologies) v2.3.4 with the '–guppy-config res_dna_r941_prom_modbases_5mC_v001.cfg – outputs mod_basecalls –mod-motif m CG 0 –devices cuda:0 –processes 48 –overwrite'.

## HiPore-C data alignment pipeline

The HiPore-C alignment analysis pipeline requires using ngmlr v0.2.7[60] and minimap2 v2.17-r941[61] software. Reads were first aligned to the reference genome (GRCh38) using ngmlr with the parameter "–subread -length 256 -x ont" and minimap2 paftools.js sam2paf to convert from sam format to paf format. In the preliminary alignment, unaligned reads were realigned using minimap2 with the parameter "-x map-ont -B 3 -O 2 -E 5 -k13", and then the two alignment results were combined. Different parts of the reads were mapped to distinct genomic loci and called fragments. There were gap openings and overlap between fragments (Supplementary Fig. 1g). If the alignment strand and genomic position of the two overlapping fragments were coincident (the dislocated overlapping genome positions were within 50 bp), the two fragments were merged. Otherwise, the shorter alignment fragment was discarded. After processing overlapping fragments, we extracted the gap regions from the alignment reads and realigned them with the same parameters using minimap2: "-x map-ont -B 3 -E 5 -k13". The alignment fragments were annotated with the genome in silico DpnII restriction digestion fragments, and we defined fragment ends located within 30 bp of the digestion sites as the match ends. If both fragments' ends matched, the fragment was fully digested. To obtain reliable alignment results, we discarded fragments with a mapq score <10 without match ends. After annotating the multiple fragments of reads, each multi-fragment read represented a high-order chromatin interaction. For comparison with Hi-C data, the multiway contacts of HiPore-C reads were decomposed into pairwise contacts (Supplementary Fig. 1h). A read with n ligated fragments was able to generate $C(n, 2)$ pairwise contacts, and a pairwise contact matrix file was generated to juicer medium format. Pore-C datasets were analyzed in the same way.

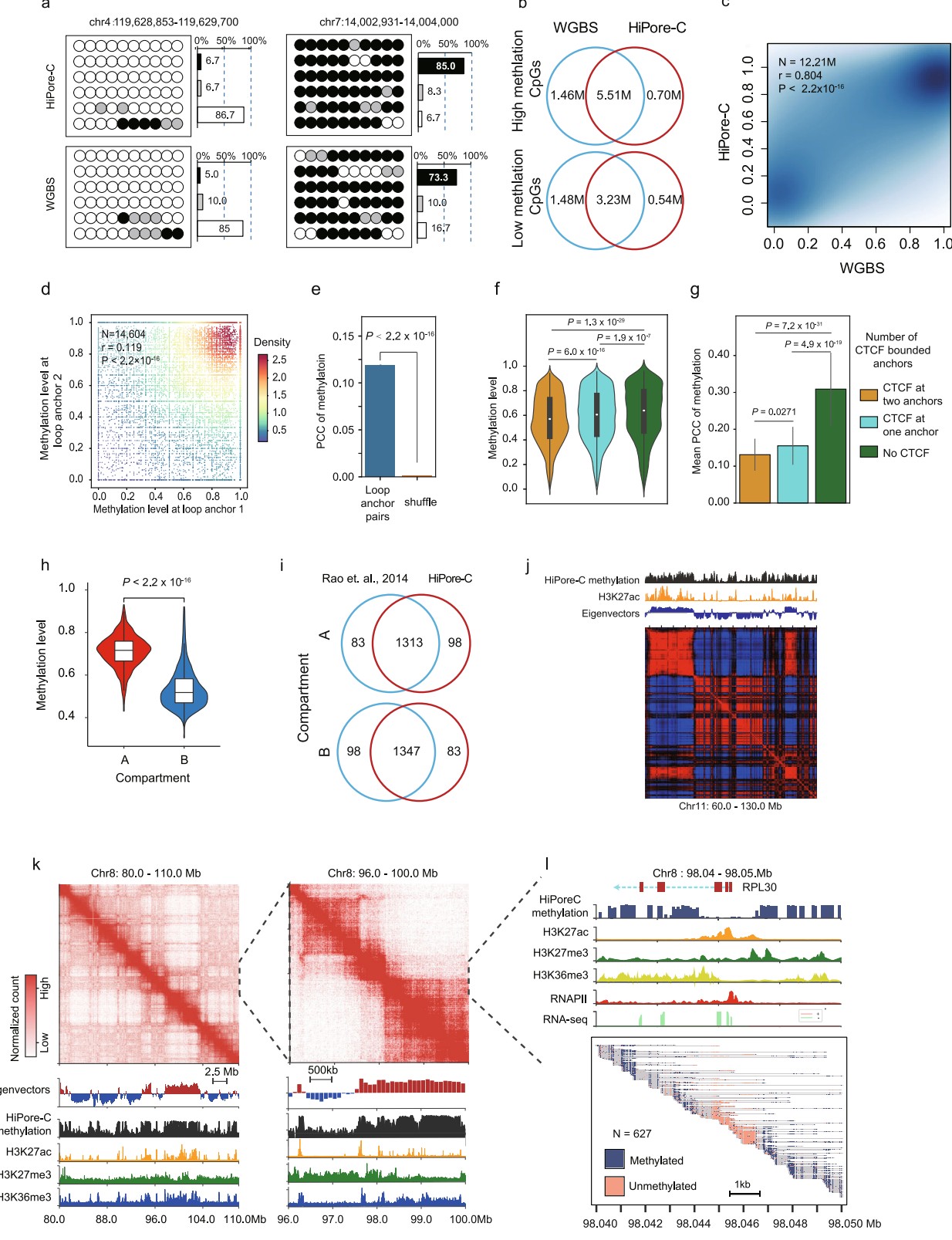

## Comparison of HiPore-C and Hi-C data

We obtained a total of 1.35 billion pair contacts from 5 runs of the GM12878 HiPore-C datasets, and we obtained the previously reported GM12878 cell in situ DpnII digestion Hi-C dataset containing 421.7 million pairwise contacts from the 4DN Data Portal (4DNESQWI9K2F) for GM12878 cell line[14] and 601.97 million pairwise contacts (4DNESF17LNZE) for K562 cell line[70]. We used cooler v0.8.6.post0[71] to

normalize the HiPore-C and Hi-C pairwise contact matrix to generate data in the cool and mcool formats with default parameters. To visualize the chromatin conformation contact heatmap, we used juice tools v1.22.1[72] to generate the hic file. To compare the degree of similarity between HiPore-C and Hi-C datasets and between different runs of HiPore-C, the stratum-adjusted correlation coefficient of the pairwise contact matrix between samples was calculated using HiCrep

**Fig. 7 | HiPore-C captures DNA methylation and chromatin topology simultaneously. a** CpG methylation captured by HiPore-C and WGBS. (Black, highly methylated, average methylation ratio >= 0.6; gray, medianly methylated, average methylation ratio 0.4 – 0.6; white, lowly methylated, average methylation ratio <= 0.4). Bar plots show the distribution of CpGs with different methylation levels. **b** Overlap of the highly and lowly methylated CpG between the HiPore-C and WGBS datasets. **c**, Pearson's correlation between HiPore-C and WGBS datasets on CpG methylation levels (with 5x coverage). **d**, **e** Pearson's correlation coefficient (PCC) values of CpG methylation levels between anchors of chromatin loops[14] (25 kb resolution bin). Only reads (n = 14,604) containing three or more CpGs on both anchor fragment were used in calculation. In **e**, paired anchor fragments in reads were shuffled (n = 14,604, shuffled anchor pairs) to calculate the expected background PCCs. Comparison of the PCCs of the expected background and the observed is shown in **d**. Statistical significance was calculated using Finsher's z (1925) in cocor package tool. Distribution of anchor methylation level (**f**) and comparison of the PCC values between paired anchors (**g**) of three loop types.

Loops with CTCF at two anchors (n = 4858); Loops with CTCT at one anchor, (n = 3432); Loops without CTCF (n = 1140). Statistical significance was calculated by the Kruskal-Wallis test, followed by Dunnett's t-test. In **f** and **h**, the center line/ dot, median; boxes, first and third quartiles; whiskers, 5th and 95th percentiles. In **g**, Data are shown as the mean ± sd. **h** Distribution of methylation level in the A (n = 1396) and B (n = 1445) compartments. Statistical significance was calculated by two-sided Student's t-test. Data are shown as in **f**. **i** Venn diagrams show the overlap between A, and between B compartments that were identified by Rao et al.[14] and by using the HiPore-C captured CpG methylation, respectively. **j** Chromatin compartments and methylation profile at Chr11:60Mb-130.0 Mb. The H3K27ac track was plotted using ENCODE H3K27ac datasets. **k**, **l**, Contact heatmaps (500 kb and 50 kb resolutions) and methylation profiles at Chr8:80.0Mb-110.0 Mb. Dashed lines show the amplified regions from the left panels. In panel **l**, methylated and unmethylated regions detected by HiPoreC in local single-allele reads are shown in the bottom frame. The ChIP-seq and RNA-seq tracks were plotted using data from ENCODE database.

v1.2.0 (scc)[73]. We used eigs-cis from cooltools v0.5.0[74] to calculate compartment eigenvectors with a bin resolution of 100 kb and determined the types of compartments A and B using ENCODE GM12878 H3K27ac ChIP-Seq data (ENCFF798KYP). We used cooltools v0.5.0 to calculate insulation scores for TAD at 50 kb resolution and window sizes of 2, 5, and 10. We also separately calculated the Pearson's correlation r of compartment eigenvector scores and TAD insulation scores between the two methods. We used the juicer apa tool to compare the results of aggregate peak analysis (APA) for the loops between HiPore-C and Hi-C datasets and loops derived from a previous study[14].

### Analysis of 3D genome high-order interactions

**Analysis of multiway contacts.** Previous studies have reported that multiway contact reads can capture longer-range genomic interactions than Hi-C-captured pairwise contacts[57]. We calculated three types of contact distances in terms of the relative locations of fragments in HiPore-C reads, which were read cover distance (the maximum genomic distance covered by a read), adjacent contact distance (distance between pairs of adjacent ligation fragments), and separated contact distance (distance between pairs of separated fragments) (Fig. 4a), and compared them with the Hi-C pairwise-contact distance. The contact distances of HiPore-C reads with different numbers of fragments, lwLRMFs (2-3), mdLRMFs (4-9), and hgLRMFs (>=10), were also analyzed, where lw indicates low, md indicates medium, and hg indicates high. We collected loops, TADs, and compartment information of the GM12878 cell line (GSE63525)[14]. We analyzed reads spanning multiple chromatin structural domains (loop anchors and regions, TADs, and compartments) with different numbers of fragments.

**Comparison of adj-pairs and non-adj-pairs of chromatin contacts in multi-contact reads.** When generating pairwise contacts, we separated contacts between two neighboring fragments (adj-pairs) from the rest (non-adj-pairs) and generated contact matrices for these two types of chromatin interaction pairs separately. The matrices of the non-adj-pairs and Hi-C contained more contacts than the matrix of adj-pairs. We down-sampled Hi-C and non-adj-pairs datasets to the same amount of the adj-pairs by cooltools random-sample. Then, we calculated the stratum-adjusted correlation coefficient, compartment eigenvectors and insulation scores, and the aggregate peak analysis. Finally, to test whether inter-chromosomes, inter-compartments, and inter-TADs contacts (named inter-contacts) were enriched in the adj-/non-adj-pairs datasets, we set the proportion of the inter-domain contacts in all pairwise contacts as the expected ratio. We then multiplied it by the number of adj-/non-adj-pairs to derive the expected values. The enrichment score was calculated by dividing the observed inter-contact number by the expected value. The enrichment scores of

intra-chromosome, intra-compartment, and intra-TAD contacts were similarly calculated.

**Hierarchical clustering of multiway contacts.** Taking advantage of the informative multiway interactions within the HiPore-C reads, we analyzed differences between single-allele topologies to improve our understanding of the cell type-specific chromatin conformations. We performed hierarchical clustering on high-order reads in a specific region to study the chromatin interaction complexity in TAD regions. To facilitate the observation of long-range multiway interactions, we selected reads containing more than four fragments in specific regions to cluster. According to its size, the region of interest was divided into M bins (for example, 1 kb bins if the region was less than 200 kb, otherwise 5 kb), and N (number) read fragments were assigned to corresponding bins if the fragment midpoint fell within a bin. If $bin_j$ was in read i, then $P_{i,j}$ is 0; otherwise, 0. This resulted in a $P[N \times M]$ matrix containing reads in the rows and region bins in the columns. We used the Python scipy package for hierarchical clustering (scipy.cluster.hierarchy) with the matrix distance generated by "euclidean" and the clusters generated by the "ward" method, and branch distance was adjusted to achieve read hierarchical clustering in this region. The relative frequency for each cluster bin was calculated as the observed frequency of every bin divided by the number of reads in each cluster:

$$Relative\, freq = bin\, number / reads\, number \qquad (1)$$

To analyze the profile of multiway contact of CTCF sites, we obtained CTCF peaks in GM12878 cells (ENCFF796WRU) from ENCODE and the CTCF motif weight matrix from JASPAR. MEME Suite FIMO software[75] was used to identify motifs in regions of CTCF binding peaks with a p-value threshold of $1e^{-4}$. We calculated each cluster's relative frequencies of CTCF fragments (fragments located in CTCF regions) and CTCF pairwise contacts (pairwise contact fragments located in CTCF regions).

To visualize the contact heatmap of each cluster, we created a pairwise contact matrix for each cluster of reads, and normalization and visualization were conducted as described above. To compare the interaction distances of different cluster reads, we calculated the distance observed/expected (O/E) by taking the cover distance of reads or pairwise contact distance of all the reads as expected values (E) and contact distance in the cluster reads as the observed values (O).

**Multiway contact of regulatory elements.** To investigate the heterogeneity of high-order interactions at the human β-globin gene locus, we performed hierarchical clustering of multiway contact reads in GM12878 and K562 cell lines as mentioned above. The relative association frequency between the gene regulatory region of interest (X) and other regulatory regions was calculated in each clustered read.

We repeated sampling in each cluster, and if X preferentially interacted with regulated targets, it was also present at high frequency in the sampling datasets. We sampled 100 times, calculated the frequencies of targets in reads containing X elements in each sampling dataset, and divided them by the number of subsets reads for normalization. To analyze the differences in simultaneous interactions between multiple promoters and multiple enhancers among different clusters, we calculated and compared the frequencies of reads containing three-way interactions in the regulatory regions of interest in the sampled datasets among different clusters. The means and standard deviations of relative frequencies were calculated, and the significance of differences between clusters was calculated using Welch's t-test and Bonferroni's multiple test correction, with alpha = 0.01.

**Multiway promoter and enhancer interaction analysis.** To analyze the global multiway interaction of cis-regulatory elements, we adopted the multi-promoter interaction model[27] and set up a multi-enhancer interaction model (Supplementary Fig. 8a). We obtained the V15 ChromHMM annotations of GM12878 and K562 cell lines from the hg19 ENCODE data resource (http://genome.ucsc.edu/ENCODE/downloads.html). The annotations were lifted to the reference hg38 genome via the liftOver utility tool from the University of California Santa Cruz. We then selected 'strong enhancers' and 'active promoters' for further analysis. In addition, the promoter needed to be located within 2 kb upstream of the gene TSSs in the Encode GRch38 V29 genome annotation (https://www.encodeproject.org/data-standards/reference-sequences). The promoter and enhancer regions were binned in 2 kb resolution, and the multiway contacts were counted in each bin. Some of the promoter bins and enhancer bins overlapped. In the promoter interaction model, the overlapped bins were all treated as promoter bins, while in the enhancer interaction model, they were treated as enhancer bins. In the promoter interaction model, a basal promoter (BP) read contains only one promoter fragment and no enhancer fragment; a single-gene (SG) interaction read contains only one promoter fragment and one or more enhancer fragments; a multi-gene interaction (MG) read contains two or more promoter fragments. In the enhancer interaction model, a none-enhancer interaction (NE) read contains only one promoter fragment and no enhancer fragment; a single-Enhancer interaction (SE) read contains at least one promoter fragment and only one enhancer fragment; a multi-enhancer interaction (ME) read contains at least one promoter fragment and two or more enhancer fragments. Then, we calculated the frequency of distinct interaction for those gene which are covered by promoter fragment (Supplementary Fig. 8a).

To analyze the association of multiway interaction of cis-regulatory elements with gene expression, we obtained RNA-seq data of GM12878 (ENCFF678BLG, ENCFF897XES, ENCFF791MED, ENCFF473KMX) and K562 (ENCFF068NRZ, ENCFF928YLB, ENCFF472HFI, ENCFF628SMT) from the Encode database. We normalized gene expression level as the mean value of transcripts per million (TMP). We divided genes into groups with the lowest interaction frequency (Q1, <25% interaction frequency), moderate interaction frequency (Q2, ≥25% and <75% interaction frequency), and the highest interaction frequency (Q3, ≥75% interaction frequency).

**Analysis of interchromosome interactions**
**Identification of interchromosomal interactions.** We divided chromosomes into 1 Mb bins and converted the interchromosomal interactions in the multiway contact reads into a pairwise contact matrix. According to the reported method, the significance of interchromosomal interaction enrichment was calculated using the negative binomial distribution with Bonferroni's multiple corrections based on the assumption that interchromosomal interactions were randomly distributed. We selected significantly enriched

interchromosomal interactions by an enrichment score >= 2 and adjusted p-value <0.01 based on the distribution profile of enrichment scores and adjusted p-values, and then we selected contact pair ij with two other consecutive bins that were significantly enriched contact pairs (i.e., the i+1 and j+1 and the i-1 and j-1 contact pairs were significantly enriched). To exclude false-positive interaction bins further, we required that the enriched bins have interactions with multiple regions (at least 20 other bins). Finally, 623 bin regions were identified as significant interchromosomal interacting loci. Enrichment analysis was performed for the interchromosome interactions with the centromere, telomere, and tRNA genes in the anchor regions.

**Interchromosomal interaction hubs**
It was reported that two classes of interchromosome interaction hubs could be identified from multiple contacts[6]. We transformed the interactions of the 623 regions into a 623*623 matrix with M i, j = 1 if there were significantly enriched interactions in regions i and j; otherwise, 0. We then used the Gaussian mixture model from the Python sklearn library, taking the matrix as input, to partition these regions into two sets. In each set, we selected regions with a significant degree of connectivity within the same set and a small degree of connectivity with the other set (regions with a contact ratio within the same set ≥ 0.9). We obtained two interchromosome interaction hubs, which contained 72 and 78 regions. We analyzed the features of genomic regions of these two hubs, including epigenetic histone modifications (H3K4me1: ENCFF321BVG; H3K4me3: ENCFF587DVA; H3K27ac: ENCFF023LTU; and H3K36me3: ENCFF432EMI), RNA polymerase II (RNAPII) ChIP-seq occupancy(ENCFF916VXY), and DNase I hypersensitivity (ENCFF759OLD), as well as the densities of genes and enhancers. The histone modifications, RNAPII ChIP-seq occupancy, and DNase sensitivity were defined as the number of peaks (ENCODE) per Mb region, and densities of the gene (Ensemble gene annotation) and enhancer (ENCODE candidate enhancers ENCFF733BFV) were defined as the counts of genes and enhancers per Mb region, respectively. The RNA expression level was defined as the mean total RNA-seq fold change over the control level per Mb region. The average value of these features in two hub regions was calculated, and the interchromosomal contact-enriched regions not from the two hubs were used as the control group. One hub was considered an active hub because of the higher genomic accessibility and histone modifications related to its active transcription state. The other hub was considered a transcriptionally inactive hub.

**Analysis of HiPore-C methylation**
**Comparison of HiPore-C methylation with the conventional method.** We extracted 5mC methylation sites from the HiPore-C dataset of megalodon bam files using a customized Python script and set a methylation probability score greater than 191 (i.e., methylation possibility greater than 0.75) as the threshold for methylation C base calling. The megalodon bam files were converted to fastq files, and alignment and annotation were performed as described in the HiPore-C data alignment pipeline. The CpG sites were mapped to the reference genome according to read annotations. There were 2.53 billion CpG methylation calls and 1.40 billion CpG unmethylation calls (Supplementary Table 7).

To evaluate the reliability of the HiPore-C methylation results, we calculated the methylation ratio of CpG sites in the reference genome using the WGBS dataset as a control (100× coverage of GM12878 WGBS data, ENCODE accession number ENCFF067JYV). The Pearson correlation coefficient for CpG methylation between the WGBS and HiPore-C datasets was calculated. The concordance of highly methylated CpG sites (methylation ratio >= 0.6) and lowly methylated CpG sites (methylation ratio <= 0.4) between these two methods was also calculated.

To determine the GC density bias in HiPore-C methylation calling, we used 1 kb bins. We analyzed the coverage of genome bin regions against the GC percentage by fitting a linear regression model of GC percentage -log10(bin count) and using the slope of the fitted straight line to reflect GC density bias.

**Association between CpG methylation and 3D chromatin structure.** We analyzed the CpG methylation profile associated with chromatin structures in the GM12878 cell line. Reads containing fragments at loop anchors and in compartments (GSE63525)[14] were kept. For reads with paired fragments in loop anchor regions (at least 3 CpG sites in the loop anchor regions), the average CpG methylation level of each read fragment of the pair of contact fragments was calculated, and the Pearson correlation coefficient (PCC) was calculated from the average methylation levels of paired fragments. To compare the difference between the background expected PCC and the observed PCC, we selected reads containing paired anchor fragments and shuffled these fragments between reads, then subjected them to PCC calculation. Correlation comparing was performed using Finsher's z (1925) in cocor tool[76] (v1.1-3, http://comparingcorrelations.org).

We classified loops into three groups (anchors at both ends with CTCF binding, only one anchor with CTCF binding, and neither anchor with CTCF) using CTCF ChIP-Seq data (CTCF narrow peaks, ENCODE accession ENCFF796WRU) and compared the methylation levels and calculated correlation coefficients between the two ends. We also divided loops into high- and low-level groups (top 10% vs. bottom 10% ranked by the peak density and average signal levels in loop regions) according to the DNAse-seq (ENCODE accession ENCFF960FMM), H3K27ac ChIP-seq (ENCODE accession ENCFF469WVA), and RNA-seq (ENCODE accession ENCFF936ZZD and ENCFF808QGQ) datasets and compared the methylation levels and correlation coefficients of the high- and low-level groups.

We determined the methylation difference of A/B compartments by calculating the average CpG methylation level in the compartments. To assess whether the HiPore-C methylation results could be used directly to classify A/B compartments, we performed a simple classification of A compartments ($n = 1396$) and B compartments ($n = 1445$) of the GM12878 cell line (GSE63525)[14] with two rules: higher methylation levels in the A compartment than in the B compartment and significant methylation changes between adjacent A and B compartments. We also analyzed CpG methylation in the gene promoter region at the singl-allele level using R scripts from a previous study[77].

**Quantification, statistical, and visualization.** Plots and statistics were generated in Python 3.7, R version 3.3.1, and Microsoft Excel 2016. All P values and Pearson correlation coefficients, the exact values of the numbers, and each applied statistical test are specified in the figure or figure legends. The bar graphs show the mean ± standard deviation (SD), as indicated in the figure legends. To compare two different groups, we applied a two-sided Welch t-test, and a Bonferroni−Holm correction was used to avoid errors in cases of multiple testing. To compare more than two groups, we applied the Kruskal−Wallis test, followed by Dunnett's t-test. The results were significant when $P < 0.05$ for the respective statistical test, with significance as $*P < 0.05$, $**P < 0.01$, and $***P < 0.001$.

The Juicebox (v2.10.01)[72], HiCExplorer (3.6)[78], HiGlass(v1.11.7)[79], and FAN-C(v0.9.23)[80] were utilized for depicting contact matrices and interactions, respectively.

**Reporting summary**
Further information on research design is available in the Nature Portfolio Reporting Summary linked to this article.

## Data availability
The data that support this study are available from the corresponding authors upon reasonable request. The HiPore-C sequencing data generated in this study have been deposited in the NCBI GEO database under series accession number GSE202539. The processed data are available at http://www.tgsbioinformatics.com/HiPore-C. Publicly available sequencing datasets analyzed in this study are as follows: GM12878 Hi-C data (4DNESQWI9K2F). ChIP-seq datasets include H3K27ac (ENCFF798KYP), CTCF (ENCFF796WRU), H3K4me1 (ENCFF321BVG), H3K4me3 (ENCFF587DVA), H3K27ac (ENCFF023LTU), H3K27ac (ENCFF469WVA), H3K36me3 (ENCFF432EMI), and RNA-PII (ENCFF916VXY). DNase I hypersensitivity (ENCFF759OLD) and DNAse-seq (ENCODE accession ENCFF960FMM). GM12878 WGBS (ENCFF067JYV). RNA-seq datasets (ENCFF678BLG, ENCFF897XES, ENCFF791MED, ENCFF473KMX, ENCFF068NRZ, ENCFF928YLB], ENCFF472HFI, ENCFF628SMT, ENCFF936ZZD and ENCFF808QGQ).

## Code availability
The custom Python and shell scripts used in this project are available on GitHub (https://github.com/zhengdafangyuan/HiPore-C).

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

## Acknowledgements

We thank all those who generated and freely released the data analyzed in our present study. We acknowledge financial support from the National Key R&D Program of China (2022YFF1201900 to C.X.), the National Natural Science Foundation of China (no. 91953122, 32270713, 31871326, 62150048, to C.X., and no. 32100522 to J.Z.), the Local Innovative and Research Teams Project of Guangdong Pearl River Talents Program (no. 2017BT01S138 to C.X.), CAMS Innovation Fund for Medical Sciences (no. 2019-I2M-5-005 to C.H.), Shenzhen Fundamental Research Program (no. JCYJ20220531091611025 to L.N.) and the Shenzhen Science and Technology Innovation Commission (no. 20200925153547003 to C.H.).

## Author contributions

C.X. and C.H. conceived the study. L.N. performed the experiments. J.Z. carried out data analysis. Z.L., X.B., and Y.C. contributed to the data analysis. C.X. and C.H. supervised experiments and data analysis. C.H. and J.Z. wrote the manuscript with input from all authors. F.L. contributed to the drafting of the manuscript.

## Competing interests

The authors declare no competing interests.
