## [Peer Review File · Nature Communications]

REVIEWER COMMENTS

Reviewer #1 (Remarks to the Author):

In this manuscript, Zhong, Niu, and colleagues present High-throughput Pore-C (HiPore-C), a genome-wide multi-way chromosome conformation capture (3C) approach. In contrast to conventional 3C methods such as Hi-C, multi-way 3C approaches have the advantage that they can capture multiple chromatin interactions in a single sequence read and thus provide additional information about higher-order chromatin structures formed at individual alleles. The HiPore-C approach is based on the recently developed Pore-C approach (Deshpande et al. Nature Biotechnology 2022). The authors present some technical modifications to the procedure that prevent clogging of Nanopore sequencing devices and thereby increase the throughput of the procedure. The authors show that processing HiPore-C data as conventional pairwise 3C data recapitulates structures detected by Hi-C, thus confirming that the HiPore-C approach does not introduce bias in the data. The authors next use HiPore-C to get more insight into the kind of multiway structures that are formed and investigate the extent to which such topologies span multiple chromosomes, compartments, and TADs. Finally, the authors show that HiPore-C can also be used to analyze DNA methylation patterns.

Comments

1. Even though the modifications made to the Pore-C procedure are minor (introducing additional protein digestion with Protease K and Pronase, and optimizing time and temperature), the optimized HiPore-C procedure seems to generate much more output at lower costs. However, Table S1 suggests that part of the improvement is due to the improved quality of the Nanopore flow cells and not directly related to the improved HiPore-C workflow. It would be helpful if the authors would acknowledge this in the main text of the paper and quantify the proportion of the improved outcome that results from Nanopore flow cell quality and the proportion that results from their technical modifications to the protocol. In addition to a higher sequencing output, the authors also show in Figure 1h that HiPore-C reads on average contain many more contacts compared to Pore-C reads. This is great, but it is not clear to me how increased protein digestion would result into the sequencing of longer reads with more ligation junctions. I wonder if the authors could comment on this? Finally, I wonder why the authors used DpNII and not NlaIII (as used in Pore-C) for chromatin digestion. As NlaIII on average generates smaller fragments, this would allow for more ligation junctions in a single sequencing read.
2. The authors use their HiPore-C data to investigate how multi-way interactions are formed across chromosomes, compartments, and TADs. It is important to note that such analyses are not completely novel. Since the HiPore-C data appear to be of high quality, I think these analyses are still worthwhile and interesting, but I do think it would be good if the authors would relate their findings more clearly to the current literature. For example, the question of how multiway contacts span chromosomes, compartments and TADs has previously been addressed by Olivares-Chauvet et al (Nature 2016), but the authors do not relate their findings to this work. In addition, the tissue-specific nature of multiway contacts at the beta-globin locus (in mouse) has been studied by Allahyar et al (Nature Genetics 2018) and Oudelaar et al (Nature Genetics 2018). The authors do cite these papers, but not in this context.
3. Figure 4b-d and Extended Data Figure 4: The analyses presented in these figures suggest that non-adjacent fragments in long sequencing reads have a different distance decay curve compared to pair-wise interactions in Hi-C data or adjacent fragments in long sequencing reads. I find this an interesting observation, which I think would benefit from more

discussion as this could have important implications. I wonder if it is valid to consider restriction fragments that are separated by many other fragments in a single sequencing read as a "contact"?

4. The authors use the term "single-allele topology polymorphism". I find this a bit confusing, as the word polymorphism in this context would imply that the authors use genetic polymorphisms to solve single allele structures, whereas this is actually not the case.

5. Figure 6-d and Extended Data Figure 6e-h: These panels show 1D interaction profiles from a single viewpoint. I do not understand how the multi-way nature of contacts is represented in these figures. To my knowledge, one would either need 2 anchors and an 1D profile (as used by Allahyar et al. Nature Genetics 2018) or 1 anchor and a 2D matrix (as used by Oudelaar et al. Nature Genetics 2018).

Reviewer #2 (Remarks to the Author):

This manuscript describes the optimization of long-read HiC using two different human cell lines and a bioinformatics pipeline to analyze multi-contact HiC data. Using this data, the authors then take three different examples of interchromosomal clustering, loops, TADs and enhancers to demonstrate the usefulness of the method. In particular, the github repository is well documented (although the link in the review is not working, but the link in <http://www.tgsbioinformatics.com/HiPore-C/> is). The paper is well written, easy to understand, yet lacks some experimental/data analysis details which would help understanding better what kind of conclusions can be extracted from multicontact HiC (see below). Nevertheless, I believe the (easy for others to apply) optimization described in the manuscript, as well as the detailed description of a multicontact HiC dataset should get published, pertaining some additional data analysis.

Major comments.

For each analysis of TADs, loops and enhancers, the authors focus on one example. While this is certainly a good start, one would like to know how the findings can be generalized. For example, would one find the same when 10 different regions and the contacts between TADs are analyzed (Fig. 4)? Similarly, does the analysis stands for 10 different loop anchors? Other differences in enhancers/promoters contacts were described in the two cell lines analyzed, how do these contact each other in the multicontact HiC dataset?

Minor comments.

- The manuscript would profit from a more detailed description of the genomic intervals analyzed, and more generally a statement or a metric of the maximum resolution possibly achieved by the technique using the current dataset. For example, what is the bin size of the loop anchors described in Figure 4f?
- It remains unclear to me how contacts are counted. If a read contains sequence A, B and C (in this order), do the authors count contacts as A/B and B/C or do they consider A/C as a contact too?
- For figure 3, how does the interchromosomal clustering correlate with A/B compartments?
- For the analysis of telomeres/centromeres (p15), how does the contact frequency observed correlate with the size of the regions analyzed? Is this proportional or is there a higher contact frequency within/between these regions than expected from regions sizes?
- Which size are tRNA bins?
- Page 20, please describe in more details what is meant by loop-string-loop structures? Page 25, it is not clear to me what the "curved dumb-bell like structure" refers

to.

- Page 26, how does the structure observed at the globin locus compare to the ones previously described using MC4C?

Response to reviewers

To both reviewers:

We thank you very much for your critical comments. We have attempted to address all of them and believe this has greatly improved our manuscript. We address your comments below point-by-point.

Reviewer #1 (Remarks to the Author):

In this manuscript, Zhong, Niu, and colleagues present High-throughput Pore-C (HiPore-C), a genome-wide multi-way chromosome conformation capture (3C) approach. In contrast to conventional 3C methods such as Hi-C, multi-way 3C approaches have the advantage that they can capture multiple chromatin interactions in a single sequence read and thus provide additional information about higher-order chromatin structures formed at individual alleles. The HiPore-C approach is based on the recently developed Pore-C approach (Deshpande et al. Nature Biotechnology 2022). The authors present some technical modifications to the procedure that prevent clogging of Nanopore sequencing devices and thereby increase the throughput of the procedure. The authors show that processing HiPore-C data as conventional pairwise 3C data recapitulates structures detected by Hi-C, thus confirming that the HiPore-C approach does not introduce bias in the data. The authors next use HiPore-C to get more insight

into the kind of multiway structures that are formed and investigate the extent to which such topologies span multiple chromosomes, compartments, and TADs. Finally, the authors show that HiPore-C can also be used to analyze DNA methylation patterns.

Comments

1. Even though the modifications made to the Pore-C procedure are minor (introducing additional protein digestion with Protease K and Pronase, and optimizing time and temperature), the optimized HiPore-C procedure seems to generate much more output at lower costs. However, Table S1 suggests that part of the improvement is due to the improved quality of the Nanopore flow cells and not directly related to the improved HiPore-C workflow. It would be helpful if the authors would acknowledge this in the main text of the paper and quantify the proportion of the improved outcome that results from Nanopore flow cell quality and the proportion that results from their technical modifications to the protocol.

Response: We thank the reviewer for the valuable suggestions. We agree with the reviewer that the improvement in the output could be caused by both the methodological optimization and the improved quality of the Nanopore flow cells.

To address the reviewer's comment on the output improvement from Nanopore flow cell quality, we carried out Pore-C experiments¹ on GM12878 and K562 cell lines, sequenced the libraries on the PromethION platform, and compared them to the datasets generated on the PromethION platform from the study of Deshpande et al.¹. Comparison analysis shows a 60% improvement in sequencing output that results from the improvement in Nanopore flow cell quality. Please see Figures 1a and Supplementary Table 1. We added a sentence in the revised manuscript acknowledging the effects of nanopore flow cell quality on the sequencing output. The sentence reads: "Although an average 60% increase in sequencing output resulted from the improved flow cell quality, the Pore-C sequencing output is well below the whole genome sequencing." Line 83-85.

To address the reviewer's comment on the output improvement from our modifications to the protocol, we compared the sequencing outputs of HiPore-C and Pore-C conducted by us. As shown in the revised Figures 1a and 1e, despite the 60% increase in the Pore-C sequencing output resulting from the improvement in Nanopore flow cell quality (as mentioned above), our HiPore-C sequencing output is about 1.8 folds (1.7-1.9 folds from different cell lines) higher than our Pore-C experiments. This additional increase in sequencing output must result from optimizing the library preparation protocol in our HiPore-C experiments. We state this improvement in line 112-113, which reads, "further improved the sequencing yield by about 80% compared to Pore-C (Fig. 1e)."

In addition to a higher sequencing output, the authors also show in Figure 1h that HiPore-C reads on average contain many more contacts compared to Pore-C reads. This is great, but it is not clear to me how increased protein digestion would result into the sequencing of longer reads with more ligation junctions. I wonder if the authors could comment on this?

Response: This is a great question.

The number of contacts in Pore-C and HiPore-C reads largely depends on the length of the sequenced reads and the size of the fragments digested with a particular restriction enzyme. Deshpande et al. used two restriction enzymes (DpnII and NlaIII) in their Pore-C experiments. We used only DpnII in our HiPore-Cs. So when comparing the contact numbers in reads, we can only use Deshpande et al.'s DpnII datasets. The average length of the sequenced reads in their DpnII library is only 2kb.

In contrast, the read lengths of our HiPore-C and our new Pore-C libraries are close to 5kb (4.71kb for Pore-C, 4.94kb for HiPore-C). However, the sizes of DpnII fragments in their Pore-C and our Pore-C and HiPore-C libraries are similar (733 bp, 739bp, and 709bp, respectively). So, we reasoned that the lower number of contacts in Deshpande

et al.'s Pore-C is caused by the short length of the sequenced reads. We added our Pore-C contact statistic results in Figure 1h and showed that the contact number distribution pattern of Pore-C is similar to HiPore-C.

Finally, I wonder why the authors used DpnII and not NlaIII (as used in Pore-C) for chromatin digestion. As NlaIII on average generates smaller fragments, this would allow for more ligation junctions in a single sequencing read.

Response: Indeed, as the reviewer notes, generating smaller fragments allows for more ligation junctions in a single sequencing read. We used DpnII at the beginning of our HiPore-C optimization process. To keep this critical condition comparable, we did not try other enzymes.

Through private communications, we learned that another team is testing DNaseI digestion and using our HiPore-C optimized de-crosslinking and protein removal protocol to acquire an even higher resolution of high-order chromatin interactions.

In the future, we will use NlaIII, and perhaps also DNaseI, to achieve higher resolutions.

2. The authors use their HiPore-C data to investigate how multi-way interactions are formed across chromosomes, compartments, and TADs. It is important to note that such analyses are not completely novel. Since the HiPore-C data appear to be of high quality, I think these analyses are still worthwhile and interesting, but I do think it would be good if the authors would relate their findings more clearly to the current literature. For example, the question of how multiway contacts span chromosomes, compartments and TADs has previously been addressed by Olivares-Chauvet et al (Nature 2016), but the authors do not relate their findings to this work. In addition, the tissue-specific nature of multiway contacts at the beta-globin locus (in mouse) has been studied by Allahyar et al (Nature Genetics 2018) and Oudelaar et al (Nature Genetics 2018). The authors do cite these papers, but not in this context.

Response: We thank the reviewer for pointing out that we did not relate our findings more clearly to the current literature. We have tried to relate our HiPore-C work to other publications from several aspects in the revised manuscript.

First, our HiPore-C reveals abundant inter-chromosome interactions (about 37% of reads span multiple chromosomes, Figure 3a), showing a positive correlation between the number of contacts in reads and the number of chromosomes covered, consistent with the study of Olivares-Chauvet et al.². In addition, we also found inter-chromosome interactions happen more frequently between relatively smaller chromosomes (Olivares-Chauvet et al., 2016, Nature; Tavares-Cadete et al., 2020, Nat Struct Mol Biol)^{2,3}.

Moreover, our HiPore-C reveals two interaction hubs with different transcriptional activities (Quinodoz et al., 2018, Cell)⁴.

Second, multiplex chromatin interactions were reported between compartments (A-A, B-B, A-B) (Tavares-Cadete et al., 2020, Nat Struct Mol Biol; Quinodoz et al., 2018, Cell)^{3,4}, TADs (Olivares-Chauvet et al., 2016, Nature; Meizhen Zheng et al., 2019, Nature)^{2,5}, and loop anchors (Quinodoz et al., 2018, Cell; Meizhen Zheng et al., 2019, Nature)^{4,5}.

Besides the same phenomena observed in our HiPore-C results, our data allow the quantification of diverse types of multiplex interactions at the single allele level. However, to fully address how these multiplex interactions form, it requires integrating other types of data (transcription factor distribution, epigenome datasets, and so on) to explore the underlying mechanisms that are still intriguing.

Third, previous studies (Allahyar et al., 2018, Nature Genetics; Oudelaar, et al., 2018, Nature Genetics)^{6,7} revealed multiple promoters and enhancers of the beta-globin locus interact simultaneously to form a transcriptional regulation hub. We have not only recapitulated a similar chromatin interaction hub (Figure 6) but also revealed this hub existing at low frequency, reflecting a likely active and very dynamic balancing process between the formation and dissociation of the chromatin interaction hubs as being observed in another locus (Gabriele et al. Science,2022)⁸. Our results thus highlight that higher-order chromatin interaction patterns can be quite heterogeneous, and each particular structure could exist in only a small subset of cells.

We added a few lines relating our work to the current literature in the revised manuscript in lines 173-176, 186, 190-191, 243-244, 265-267, 274, 284-286, and 360-362.

3. Figure 4b-d and Extended Data Figure 4: The analyses presented in these figures suggest that non-adjacent fragments in long sequencing reads have a different distance decay curve compared to pair-wise interactions in Hi-C data or adjacent fragments in long sequencing reads. I find this an interesting observation, which I think would benefit from more discussion as this could have important implications. I wonder if it is valid to consider restriction fragments that are separated by many other fragments in a single sequencing read as a "contact"?

Response: The reviewer asks a critical question whether it is valid to consider two indirectly ligated fragments as a "contact."

By comparing the heatmaps generated with adj-pairs (directly ligated pair of fragments in a read) and non-adj-pairs (indirectly ligated pair of fragments in a read) of chromatin contacts, we find the overall chromatin interaction patterns are pretty similar and also similar to the Hi-C contact heatmap (Stratum-adjusted correlation coefficients are 0.938 for adj-pairs and non-adj-pairs heatmaps, 0.808 and 0.844 for the pairs of adj-pairs and Hi-C, and non-adj-pairs and Hi-C heatmaps, respectively) (Extended Data Fig. 5a). We further compared the structures of compartments, TADs, and loops. In all cases,

structural patterns generated using adj-, non-adj-, and Hi-C datasets show strong correlations (Pearson's correlation coefficients are 0.919, 0.942, and 0.982 for eigenvector scores, and 0.677, 0.706, and 0.902 for insulation scores between the pairs of non-adj-pairs and Hi-C, adj-pairs and Hi-C, and adj-pairs and non-adj-pairs, respectively) (Extended Data Fig. 5b-c). In addition, we find some loops can be identified using adj-pairs and non-adj-pairs contacts (Extended Data Fig. 5d-e). The fact that no apparent differences were observed suggests that non-adj pairwise contacts are not fundamentally different from the classical direct adj-ligation. Thus, we believe the non-adj ligation can be considered as real chromatin "contact," at least at the resolutions we analyzed the data.

Although overall chromatin interaction patterns are similar between adj-pairs and non-adj-pairs chromatin interaction matrix, we do find that adj-pairs contacts are more enriched within a structural unit while non-adj-pairs contacts are more enriched in reads spanning multiple structural units (for adj-pairs and non-adj-pairs contacts: inter-chromosomal enrichment scores are 0.45 and 1.17; inter-compartment enrichment scores are 0.599 and 1.132 (A-A), and 0.775 and 1.073 (B-B), respectively; inter-TAD enrichment scores are 0.750 and 1.081) (Extended Data Fig. 5f-h). Overall, non-adj-pairs contacts are more enriched in reads covering multiple structural units than adj-pairs and Hi-C pairwise contacts. More importantly, the fragments seem to be arranged orderly in the sequenced long-reads supporting a previously proposed conjecture that the linked segments are not randomly distributed but comply with the chromatin extension paths like C-walks, and the fragment arrangement order could have important spatial and biological implications requiring further investigation (Tavares-Cadete et al., 2020, *Nat Struct Mol Biol*)³.

We included these results in the Extended Data Figures 5 and described them in the revised manuscript in lines 197-224.

4. The authors use the term "single-allele topology polymorphism". I find this a bit confusing, as the word polymorphism in this context would imply that the authors use genetic polymorphisms to solve single allele structures, whereas this is actually not the case.

Response: The "single-allele topology polymorphism" is borrowed from a previously reported concept of "single-allele chromatin interactions" (Oudelaar Marieke et al. 2018, *Nature Genetics*)⁷, which implies that each multi-way contact read can capture interactions occurring simultaneously at single alleles and provide insight into the patterns of interactions between multiple cis-regulatory elements in individual nuclei. Because our HiPore-C reads are sequenced without PCR amplification and reflect the original, native multiple chromatin locus interactions, therefore, each read represents the multi-way interactions in a single allele, and all the reads overlapping in a given genomic region constitute the local cell population topology polymorphism. Thus, we use the concept of "single-allele topology polymorphism" here.

5. Figure 6-d and Extended Data Figure 6e-h: These panels show 1D interaction profiles from a single viewpoint. I do not understand how the multi-way nature of contacts is represented in these figures. To my knowledge, one would either need 2 anchors and an 1D profile (as used by Allahyar et al. Nature Genetics 2018) or 1 anchor and a 2D matrix (as used by Oudelaar et al. Nature Genetics 2018).

Response: We apologize for not having explained this clearly enough.

Basically, we first carried out hierarchical clustering of the multi-way reads covering the beta-globin locus to reveal the chromatin contact heterogeneity in the cell population.

Then we generated the virtual 4C maps and analyzed which regulatory regions interact with the viewpoint. For the anchors in our 1D interaction profile (Figures 6c and 6d), the curves show the aggregated frequencies of fragments that are present in the sequenced long-reads covering the analyzed beta-globin locus. So as the reviewer noticed, these 1D profiles do not show the simultaneous interactions between multiple sites. In the final step, we statistically assessed whether several sites simultaneously co-occurring (or interacting) within sequenced long-reads is significant and calculated the co-occurrence rate of these sites.

Finally, we searched for simultaneous interactions between the viewpoint and regions interacting at significantly high frequencies in the multi-way contact reads (Figures 6e and 6g). The ratio of simultaneous, multi-point interactions in cells is small (the hub ratio of interaction in Allahyar et al. study is about 5%)⁶, reflecting that many genomic interactions might be dynamic and transient in each single allele.

Reviewer #2 (Remarks to the Author):

This manuscript describes the optimization of long-read HiC using two different human cell lines and a bioinformatics pipeline to analyze multi-contact HiC data. Using this data, the authors then take three different examples of interchromosomal clustering, loops, TADs and enhancers to demonstrate the usefulness of the method. In particular, the github repository is well documented (although the link in the review is not working, but the link in <http://www.tgsbioinformatics.com/HiPore-C/> is) .

The paper is well written, easy to understand, yet lacks some experimental/data analysis details which would help understanding better what kind of conclusions can be extracted from multicontact HiC (see below). Nevertheless, I believe the (easy for others to apply) optimization described in the manuscript, as well as the detailed description of a multicontact HiC dataset should get published, pertaining some additional data analysis.

Response: We thank the reviewer for the supportive comments. We have carried out additional analysis and revised the manuscript accordingly. It is at the beginning of using HiPore-C to explore the complex high-order chromatin interaction network. We are continuing the study of this topic in more detail.

Major comments.

For each analysis of TADs, loops and enhancers, the authors focus on one example. While this is certainly a good start, one would like to know how the findings can be generalized. For example, would one find the same when 10 different regions and the contacts between TADs are analyzed (Fig. 4)? Similarly, does the analysis stands for 10 different loop anchors? Other differences in enhancers/promoters contacts were described in the two cell lines analyzed, how do these contact each other in the multicontact HiC dataset?

Response: The reviewer's suggestions are great. Due to the space limitation, we did not include more analysis or other examples in the previous version. In the revised manuscript, we include a global statistical analysis of multi-contact-containing reads that cover multi-loops, TADs, and compartments (see Extended Data Figures 4). We also added two examples of well-studied gene loci (see Extended Data Figures 9 and 10).

1. Global loop anchor analysis: Although the formation of one loop requires two anchors, the two loop anchors do not necessarily co-exist in the same read in our HiPore-C analysis because loops are identified based on pair-wise interactions derived from all contacts in HiPore-C reads. We find that 50.5% of HiPore-C reads contain one loop anchor, with 27.0% of reads containing an anchor for only one loop, and 13.5% of reads containing an anchor for multiple loops sharing the same anchor. Reads containing both loop anchors account for 3.3% of total reads, including 0.27% of total reads containing anchors for multi-loops. That 53.6% of reads contain certain anchor sequences for loops suggests that looping is a general principle of chromosome folding. At the same time, the low percentage of reads containing both anchors of a loop or anchors of multiple loops suggests that loop formation could be very dynamic, consistent with the observation in Figure 4f. We include these updated results in Extended Data Figure 4d-g and new text in line 230-241.
2. Global TAD analysis: We showed in Extended Data Fig 4e and 4g that about 54% of reads cover two or more TADs. The number of fragments in a read positively correlates with the number of TADs being covered, in agreement with the results in Figure 4g. We include these updated results in new text in line 258-261.
3. Global compartment analysis: Our statistic analysis showed that fragments in the same read (61.23% of total reads) co-localize in the same types of compartments (the same A, the same B, and two or more A or B compartments, respectively). Reads spanning both types of A and B compartments are relatively low. And

fragments in reads covering both A and B compartments still predominantly locate in one type of compartment, and these reads generally cover "multi-A-one B" or "Multi-B-one A" reads (29.26% of the total reads). Only 9.51% of reads contain fragments that show no preferential distribution in two types of compartments (adjacent A-B, non-adjacent A-B, multiple A-multiple B A+_B+ type). These results are consistent with Figure 4h. These results are shown in Extended Data Figure 4f and mentioned in the text line 284-286.

4. Global Enhancer-promoter interaction analysis: Following Li et al. 's Promoter interaction and Enhancer interaction models (Li et al.; Cell, 2013)⁹, we classified genes and enhancers based on the number of enhancers and promoters they interact with. In the Promoter interaction model, genes and enhancers are separated into three groups BP (basal promoter), SG (single-gene interaction), and MG (multi-gene interaction). In the Enhancer interaction model, genes and enhancers are separated into three groups NE (Non-enhancer interaction), SE (single-enhancer interaction), and ME (multi-enhancer interaction), as shown in Extended Data Figure 8a. Our analysis shows that many genes are involved in MG and ME interactions. And about 8.6% and 4.3% of reads are involved in MG and ME interactions, respectively (Extended Data Figure 8b). Globally, in the multiple promoter model, BP, SG, and MG interactions exist in every gene. The BP interaction is generally dominant, with an average frequency of 66.4%, while the MG frequency is the lowest, at 16.1%.
5. Similarly, in the multi-enhancer model, genes have NE, SE, and ME interactions, with the NE interactions dominating (76.7%) and a much lower portion of ME interactions (4.2%) (Extended Data Figure 8b). We also show that multiple promoter interactions do not correlate (neither positively nor negatively) with gene expression levels. In contrast, for genes associated with multiple enhancer interactions, the low-NE genes (Q1) are expressed at significantly higher levels than the high-NE genes (Q3). In addition, the levels of ME positively correlate with the levels of gene expression (Extended Data Figure 8d). In sum, multiple enhancer interactions are more significantly associated with a high gene expression level than multiple promoter interactions.
6. Analysis of two additional gene families: We add promoter and enhancer multiple interaction analysis of two well-studied gene families of the Histone gene clusters 1, 2, 3 (HIST) and the human leukocyte antigen (HLA) gene loci.

A, We observed strong interactions between promoters within the three HIST gene clusters (D1, D2, and D3) and between each pair of gene clusters (Extended Data Figure 9). The high-order interaction frequencies between promoters and enhancers are significantly higher than the genome-wide background level (Extended Data Figure 9b and 9e). Moreover, the ME-type interactions in the D1 and D2 gene clusters are different in GM12878 and K562 cells, which may contribute to the different regulation of gene expression (Extended Data Figure 9c-e).

B, We observed similar interaction patterns for the HLA gene loci in both GM12878 and K562 cell lines. Apparent heterogeneity in promoter and enhancer multi-interactions is also observed in both cell lines (Extended Data Figure 10). The proportion of ME interactions of the HLA locus is higher in GM12878 than in K562 cells, with significant differences being detected for the genes of HLA-Z, HLA-DPB1, HLA-DPA1, and HLA-DPA2. Moreover, ME intensities positively correlate with the high expression of particular HLA genes in GM12878 cells (Extended Data Figure 10 c-e).

We include the results of these new analyses in Extended Data Figures 4, 8, 9 and 10 and add text accordingly in line 230-241, 258-261, 284-286, and 369-375.

Analysis of these two additional gene families, together with our previous beta-globin locus analysis, showed that the high-order interactions among multiple promoters and enhancers are extensively established, dynamically established, and heterogeneous in cell populations.

These results highlight that HiPore-C can be used to identify and quantify the global and cell-specific high-order promoter and enhancer interactions and to reveal the interaction heterogeneity in cell populations.

Minor comments.

- The manuscript would profit from a more detailed description of the genomic intervals analyzed, and more generally a statement or a metric of the maximum resolution possibly achieved by the technique using the current dataset. For example, what is the bin size of the loop anchors described in Figure 4f?

Response: We thank the reviewer for this suggestion. The resolutions appropriate for analyzing different types of 3D genome structures vary. As suggested, we add resolutions, sizes of analyzed regions, and bin sizes in the revised Figures, Extended Data Figures, and supplemental materials.

- It remains unclear to me how contacts are counted. If a read contains sequence A, B and C (in this order), do the authors count contacts as A/B and B/C or do they consider A/C as a contact too?

Response: Yes. The reviewer is correct. The pairwise contacts include adjacent fragment pairs and fragment pairs that are not adjacent in the reads, as shown in a diagram explaining how contact pairs were derived in the Extended Data Figure 1H. Detailed explanations can also be found in the figure legend and Methods section.

Another reviewer asks a similar question about whether non-adjacent pairs can be considered "contact."

In the supplementary analysis, we found that the major differences between adj-pairs and non-adj-pairs contacts lie in 1) the interaction distances and 2) the number of structural units spanned. Non-adj contacts show longer interaction ranges and span more structural units (loop, TAD, and compartment). Nevertheless, adj-pairs and non-adj-pairs contacts can both faithfully capture the chromosome conformation features of the loop, TAD, and compartments. Therefore, adj-pairs and non-adj-pairs contacts both reflect the same genome architectures.

Our response to another reviewer's comment 3 also explains why we considered non-adjacent pair of fragments as "contact."

- For figure 3, how does the interchromosomal clustering correlate with A/B compartments?

Response: We identified two inter-chromosomal clusters 1 and 2. Cluster 1 mainly comprises chromatin in Compartment B (91.11%). In contrast, cluster 2 is composed of chromatin in Compartment A (95.01%). We include this result in the Extended Data Figure 3d and described in the text, line 173-176.

- For the analysis of telomeres/centromeres (p15), how does the contact frequency observed correlate with the size of the regions analyzed? Is this proportional or is there a higher contact frequency within/between these regions than expected from regions sizes?

Response: Qualitatively, contact frequencies between particular pairs of genomic regions are generally consistently higher or lower if compared to the genome-wide chromatin interaction background at the corresponding distances and if the resolution for analysis is within a reasonable range. By "reasonable range," we mean the bin size chosen should be appropriate for the analysis of the particular sizes of the regions, just like people use different resolutions for loops, TADs, and compartment identifications.

In the case of inter-chromosomal telomeres/centromeres interactions, we divided the genome into 1Mb bins, a size frequently used for inter-chromosomal interaction analysis. Compared to the background genomic interaction frequencies between pairs of the same-sized regions, we observed higher interaction frequencies between telomeres and pairs of centromeres (Figure 3).

- Which size are tRNA bins?

Response: Because tRNA genes are located in many chromosomes, we carried out tRNA gene interaction analysis like the inter-chromosome interaction analysis at a bin size of 1Mb. As suggested, we include the resolution for this analysis in the corresponding Figure 2b and Extended Data Figure 2e-f.

- Page 20, please describe in more details what is meant by loop-string-loop structures? Page 25, it is not clear to me what the "curved dumb-bell like structure" refers to.

Response: We apologize for not being clear. "Loop-string-loop structure" refers to a read containing two pairs of fragments forming two loops far-separated in the linear genomic distance. This structure is frequently observed in reads spanning two or more TADs.

"Curved dumbbell-like structure" refers to a particular cluster of reads that contain fragments clustered at the two ends of a TAD that are separated by the middle region in a TAD. Because these two clustered fragments are ligated in the same reads, they interact more frequently with each other than with the middle region of a TAD, like a bent dumbbell whose two ends meet.

We add a few lines (262-263 and 331-335) to explain these two structures in the revised manuscript.

- Page 26, how does the structure observed at the globin locus compare to the ones previously described using MC4C?

Response: We thank the reviewer for bringing up this important point. Our beta-globin gene locus analysis results are consistent with MC4C results⁶ in that promoters and enhancers were shown to form a hub facilitating transcription activation. Moreover, our results also reveal that enhancer-promoter multi-way contact topology is not only cell type-specific but exists in only a subset of K562 cells, suggesting the formation of this hub could be dynamic and transient in single cells. We cite the MC4C article and add a description of supplementary results in line 360-362.

Reference:

1. Deshpande, A.S. *et al.* Identifying synergistic high-order 3D chromatin conformations from genome-scale nanopore concatemer sequencing. *Nature Biotechnology* (2022).
2. Olivares-Chauvet, P. *et al.* Capturing pairwise and multi-way chromosomal conformations using chromosomal walks. *Nature* **540**, 296-300 (2016).
3. Tavares-Cadete, F., Norouzi, D., Dekker, B., Liu, Y. & Dekker, J. Multi-contact 3C reveals that the human genome during interphase is largely not entangled. *Nat Struct Mol Biol* **27**, 1105-1114 (2020).
4. Quinodoz, S.A. *et al.* Higher-Order Inter-chromosomal Hubs Shape 3D Genome Organization in the Nucleus. *Cell* **174**, 744-757 e24 (2018).
5. Zheng, M. *et al.* Multiplex chromatin interactions with single-molecule precision. *Nature* **566**, 558-562 (2019).
6. Allahyar, A. *et al.* Enhancer hubs and loop collisions identified from single-allele topologies. *Nat Genet* **50**, 1151-1160 (2018).
7. Oudelaar, A.M. *et al.* Single-allele chromatin interactions identify regulatory hubs in dynamic compartmentalized domains. *Nature Genetics* **50**, 1744-1751 (2018).
8. Gabriele, M. *et al.* Dynamics of CTCF- and cohesin-mediated chromatin looping revealed by

live-cell imaging. *Science* **376**, 496-501 (2022).

9. Li, G. *et al.* Extensive promoter-centered chromatin interactions provide a topological basis for transcription regulation. *Cell* **148**, 84-98 (2012).

REVIEWERS' COMMENTS

Reviewer #1 (Remarks to the Author):

The authors addressed most of my concerns, but there are a few remaining minor points that would be good to change:

Line 83-85: "Although an average 60% increase in sequencing output resulted from the improved flow cell quality, the Pore-C sequencing output is well below the whole genome sequencing."

It is not clear to me why the authors would compare the output to whole genome sequencing? Can the authors please clarify what they mean by this statement?

Regarding the term "single-allele topology polymorphism" – I checked, and this is not mentioned as such in the paper by Oudelaar et al. I As said, I find this term very confusing and would find the term "single-allele topology" much more intuitive.

Response to reviewers

To reviewer:

We thank the reviewer again for the critical comments. We have attempted to address them, and believe our manuscript has been further improved.

Reviewer #1 (Remarks to the Author):

Line 83-85: "Although an average 60% increase in sequencing output resulted from the improved flow cell quality, the Pore-C sequencing output is well below the whole genome sequencing."

It is not clear to me why the authors would compare the output to whole genome sequencing? Can the authors please clarify what they mean by this statement?

Response: We apologize for not making it clear.

The output of the whole genome sequencing can be considered the expected sequencing yield for native DNAs sequenced on the ONT platform. If the Pore-C/HiPore-C experimental chromatin processing causes any reduction in the sequencing output, we can find out by comparing it with the WGS. As seen in Fig. 1a, the sequencing yield of Pore-C is much lower than the WGS, suggesting a much big room for improvement in chromatin processing. We explained the purpose of this comparison in the revised manuscript. Please see lines 83-84.

Regarding the term "single-allele topology polymorphism" – I checked, and this is not mentioned as such in the paper by Oudelaar et al. I As said, I find this term very confusing and would find the term "single-allele topology" much more intuitive.

Response: We apologize for not giving a clear answer last time and thank the reviewer for the valuable suggestions.

The term "single-allele chromatin interactions" is from Oudelaar Marieke et al.^{1.}, and the term "single-allele topology" is from Allahyar et al.^{2.} We use the term "single-allele topology polymorphism" to indicate the diversity of "single-allele topology" in cell populations. Considering that this term may be confusing, we agree with the reviewer that it is more intuitive to use the term "single-allele topology." So we changed all "single-allele topology polymorphism" to "single-allele topology" in the revised manuscript.

Reference:

1. Oudelaar, A.M. et al. Single-allele chromatin interactions identify regulatory hubs in dynamic compartmentalized domains. *Nature Genetics* 50, 1744-1751 (2018).
2. Allahyar, A. et al. Enhancer hubs and loop collisions identified from single-allele topologies. *Nat Genetics* 50, 1151-1160 (2018).